# GABA signalling modulates stomatal opening to enhance plant water use efficiency and drought resilience

Bo Xu [1,2], Yu Long[1,2], Xueying Feng [1,2], Xujun Zhu [1,3], Na Sai [1,2], Larissa Chirkova[2,4], Annette Betts[5], Johannes Herrmann[6], Everard J. Edwards [5], Mamoru Okamoto [2,4], Rainer Hedrich [6] & Matthew Gilliham [1,2✉]

The non-protein amino acid γ-aminobutyric acid (GABA) has been proposed to be an ancient messenger for cellular communication conserved across biological kingdoms. GABA has well-defined signalling roles in animals; however, whilst GABA accumulates in plants under stress it has not been determined if, how, where and when GABA acts as an endogenous plant signalling molecule. Here, we establish endogenous GABA as a bona fide plant signal, acting via a mechanism not found in animals. Using *Arabidopsis thaliana*, we show guard cell GABA production is necessary and sufficient to reduce stomatal opening and transpirational water loss, which improves water use efficiency and drought tolerance, via negative regulation of a stomatal guard cell tonoplast-localised anion transporter. We find GABA modulation of stomata occurs in multiple plants, including dicot and monocot crops. This study highlights a role for GABA metabolism in fine tuning physiology and opens alternative avenues for improving plant stress resilience.

[1] Plant Transport and Signalling Lab, ARC Centre of Excellence in Plant Energy Biology, Waite Research Institute, Glen Osmond, SA, Australia. [2] School of Agriculture, Food and Wine, Waite Research Precinct, University of Adelaide, Glen Osmond, SA, Australia. [3] College of Horticulture, Nanjing Agricultural University, Nanjing, China. [4] ARC Industrial Transformation Research Hub for Wheat in a Hot and Dry Climate, Waite Research Institute, University of Adelaide, Glen Osmond, SA, Australia. [5] CSIRO Agriculture & Food, Glen Osmond, SA, Australia. [6] Institute for Molecular Plant Physiology and Biophysics, University of Würzburg, Würzburg, Germany. ✉email: matthew.gilliham@adelaide.edu.au

The regulation of stomatal pore aperture is a key determinant of plant productivity and drought resilience, and profoundly impacts climate due to its influence on global carbon and water cycling[1–3]. The stomatal pore is delineated by a guard cell pair. Fine control of ion and water movement across guard cell membranes, via transport proteins, determines cell volume and pore aperture following opening and closing signals such as light and dark[2,4,5] (Fig. 1). Due to their critical roles and their ability to respond to and integrate multiple stimuli, stomatal guard cells have become a preeminent model system for investigating plant cell signalling[6] resulting in the elucidation of many critical pathways involved in plant biotic and abiotic stress tolerance[7–9].

GABA signalling in mammals relies upon receptor-mediated polarization of neuronal cell membranes[10,11]. Speculation that GABA could be a signal in plants is decades old[12], but a definitive demonstration of its mode of action remains elusive. GABA production in plants is upregulated by stress[13,14]. It is synthesised in the cytosol via the GABA shunt pathway, bypassing two stress-inhibited reactions of the mitochondrial-based tricarboxylic acid (TCA) cycle[15,16]. GABA is therefore well known as a stress-induced plant metabolite that is fed back into the mitochondrial TCA cycle to sustain cellular energy production[12,17]. The discovery that the activity of aluminium-activated malate transporters (ALMTs) can be regulated by GABA[18] represents a plausible mechanism by which GABA signals could be transduced in plants, providing a putative—but unproven—novel signalling link between primary metabolism and physiology[19]. Stomatal guard cells contain a number of ALMTs that impact stomatal movement and transpirational water loss[20–22]. Therefore, stomatal guard cells represent an ideal system to test whether GABA signalling occurs in plants.

Significantly, here, we show that GABA does not initiate changes in stomatal pore aperture, rather it antagonises changes in pore size, which differentiates it from many of the signals known to regulate stomatal aperture[3–8]. Specifically, we find that GABA concentration increases under a water deficit and this reduces stomatal opening in an ALMT9-dependant manner. The anion channel ALMT9 is a major pathway for mediating anion uptake into the vacuole during stomatal opening[21]; GABA signal transduction via ALMT9 leads to reduced transpirational water loss, increased water use efficiency (WUE) and improved drought resilience. As such, even though guard cell signalling is relatively well defined[6,23], this study has been able to uncover another pathway regulating plant water loss. Furthermore, by revealing a mechanism by which GABA acts in stomatal guard cells, we demonstrate that GABA is a legitimate plant signalling molecule[16].

## Results

### GABA antagonises both stomatal pore opening and closure in epidermal peels, but only opening in leaf feeding experiments.
To validate whether GABA is a physiological signal that modulates stomatal pore aperture, our initial experiments used excised *Arabidopsis thaliana* epidermal peels where stomatal guard cells are directly accessible to a chemical stimuli[8,24–26]. When exogenous GABA or its analogue muscimol[14] were applied under constant light or dark conditions, neither elicited a change in stomatal aperture (Fig. 2a, b; Supplementary Fig. 1a, b). Interestingly though, we found that both compounds suppressed light-induced stomatal opening and dark-induced stomatal closure (Fig. 2a, b; Supplementary Fig. 1a, b). We then fed intact leaves with an artificial sap solution through the detached petiole with or without the addition of GABA or muscimol and examined whether this affected gas exchange rates. We found, in the GABA and muscimol fed leaves, that the increase in water loss (transpiration) stimulated by a dark-to-light transition was dampened compared to leaves fed just the artificial sap solution due to reduced stomatal conductance (Fig. 2c; Supplementary Figs. 1c, d and 2a). This is consistent with the reduced extent of stomatal opening that we observed in epidermal peels in the presence of GABA or muscimol upon a dark-to-light transition (Fig. 2b; Supplementary Fig. 1a). The gas exchange values of fed leaves were used to calculate instantaneous intrinsic WUE (iWUE) and WUE (ratios of carbon gained through photosynthesis per unit of water lost), which are key traits underpinning drought tolerance in plants[27], and both values were greater (i.e. improved) in GABA fed leaves (Fig. 2d; Supplementary Fig. 2a–c).

### GABA is a universal stomatal behaviour modifier.
To examine whether GABA or muscimol can modulate stomatal aperture beyond the response to light and dark, we examined their impact on a range of opening and closing signals using epidermal peels of *Arabidopsis*. We found both GABA and muscimol inhibited abscisic acid- (ABA, 2.5 µM) or $H_2O_2$-stimulated stomatal closure and coronatine-induced opening (Supplementary Fig. 3a, c, e, f)[8,28]. However, stomatal pores were fully closed in response to high concentrations of ABA (25 µM) (Supplementary Fig. 3b, d) or exogenous calcium in the presence of GABA or muscimol (Supplementary Fig. 3g), which indicated stomatal closure could occur in epidermal peels in the presence of GABA when the closing signal was of sufficient magnitude.

We tested whether our results could be explained by GABA or muscimol treatment permanently locking guard cells in a closed (or open) state and preventing further change in stomatal pore aperture, which would argue against GABA being a physiological signal. We did this by incubating epidermal peels in GABA or

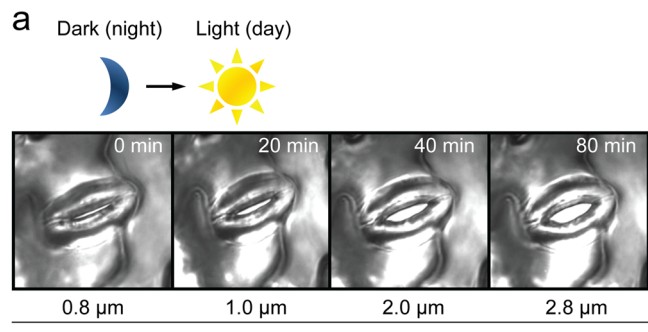

**a** Dark (night) → Light (day)

| 0 min | 20 min | 40 min | 80 min |
|---|---|---|---|
| 0.8 µm | 1.0 µm | 2.0 µm | 2.8 µm |

Stomatal aperture width

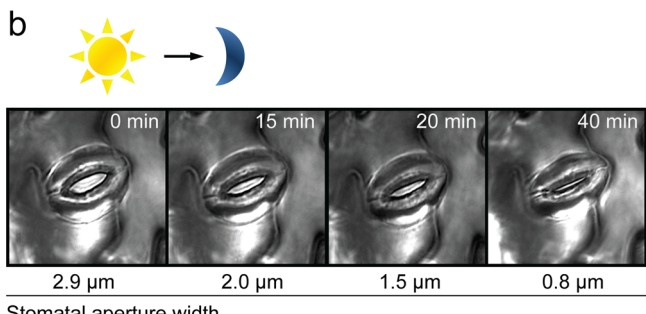

**b**

| 0 min | 15 min | 20 min | 40 min |
|---|---|---|---|
| 2.9 µm | 2.0 µm | 1.5 µm | 0.8 µm |

Stomatal aperture width

**Fig. 1 Guard cells respond to light signals. a, b** Time course of light-induced stomatal opening (**a**) and dark-induced stomatal closure (**b**) with actual stomatal aperture width indicated below; dark-to-light transition mimics night-to-day transition which opens stomatal pores (**a**) and light-to-dark transition mimics day-to-night transition which closes stomatal pores (**b**), light intensity 150 µmol m$^{-2}$ s$^{-1}$.

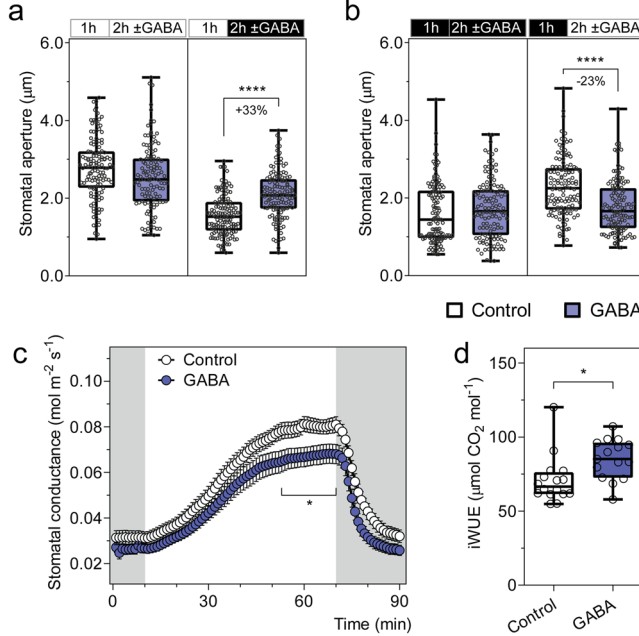

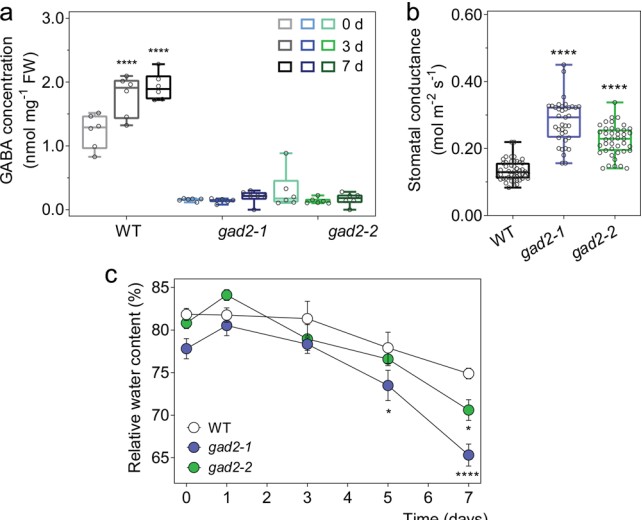

**Fig. 3 Leaf GABA concentration regulates transpiration. a** Leaf GABA concentration of 5–6-week-old *A. thaliana* wild-type (WT), *gad2-1* and *gad2-2* plants following drought treatment for 0, 3 and 7 days, $n = 6$. **b** Stomatal conductance of *Arabidopsis* WT, *gad2-1* and *gad2-2* plants determined using an AP4 porometer; $n = 48$ for WT, $n = 37$ for *gad2-1* and $n = 41$ for *gad2-2*, data collected from three independent batches of plants. **c** Relative leaf water content of WT, *gad2-1* and *gad2-2* plants following drought treatment for 0, 1, 3, 5 and 7 days, $n = 6$. All data are plotted with box and whiskers plots: whiskers plot represents minimum and maximum values, and box plot represents second quartile, median and third quartile (**a**, **b**), or data are represented as mean ± s.e.m (**c**); statistical difference was determined using two-way ANOVA (**a**, **c**) or one-way ANOVA (**b**); *$P < 0.05$ and ****$P < 0.0001$.

**Fig. 2 Exogenous GABA antagonises changes in stomatal pore aperture and increases intrinsic water use efficiency. a, b** Stomatal aperture of wild-type *A. thaliana* leaves in response to light or dark. Epidermal strips were pre-incubated in stomatal pore measurement buffer for 1 h under light (**a**) or dark (**b**), followed by a 2 h incubation under constant light (**a**), dark (**b**), light-to-dark transition (**a**) or dark-to-light transition (**b**) as indicated in the above graphs by the black (dark) or white (light) bars, together with the application of 2 mM GABA; $n = 129$ for control (constant light), $n = 121$ for GABA (constant light), $n = 137$ for control (light-to-dark transition) and $n = 135$ for GABA (light-to-dark transition) (**a**); $n = 122$ for control (constant dark), $n = 124$ for GABA (constant dark), $n = 123$ for control (dark-to-light transition) and $n = 130$ for GABA (dark-to-light transition) (**b**); all experiments were repeated twice in steady-state conditions (for both light or dark) or four times for dark-to-light or light-to-dark transitions in different batches of plants using blind treatments with similar results (**a**, **b**). GABA feeding of excised leaves reduces stomatal conductance (**c**) and increases intrinsic water use efficiency (iWUE) (**d**). **c** Stomatal conductance of detached leaves from 5- to 6-week-old *A. thaliana* wild-type plants was recorded using a LI-COR LI-6400XT in response to dark (shaded region) and 200 μmol m$^{-2}$ s$^{-1}$ light (white region), fed with artificial xylem sap solutions ± 4 mM GABA. **d** iWUE efficiency of detached leaves was calculated as the ratio of photosynthetic rate (Supplementary Fig. 2b) versus stomatal conductance (**c**); $n = 16$ independent leaves for control and $n = 15$ independent leaves for GABA, data collected from three different batches of plants (**c**, **d**). All data are plotted with box and whiskers plots: whiskers plot represents minimum and maximum values, and box plot represents second quartile, median and third quartile (**a**, **b**, **d**), or data are represented as mean ± s.e.m (**c**); statistical difference was determined by two-way ANOVA (**a**, **b**), or two-sided Student's *t* test (**c**, **d**), *$P < 0.05$ and ****$P < 0.0001$.

epidermal strips attenuated stomatal responses of other plant species to light or dark transitions, including the dicot crops *Vicia faba* (broad bean), *Glycine max* (soybean) and *Nicotiana benthamiana* (tobacco-relative) and the monocot *Hordeum vulgare* (barley) (Supplementary Fig. 5). The widespread inhibition of stomatal pore aperture changes suggests that GABA has the potential to be a universal 'brake' on stomatal movement in plants, including valuable crops.

muscimol, then removing this treatment and performing a light or dark transition. As would be expected from viable cells, after removal of the GABA or muscimol treatment, we found that stomatal guard cells responded to a light treatment by opening the pores (Supplementary Fig. 4a, b) or to a dark treatment by closing pores (Supplementary Fig. 4c, d). Collectively, these data again indicate that GABA signals would likely act to modulate stomatal aperture in the face of a stimulus rather than stimulating a transition itself.

To test whether GABA is a universal modulator of stomatal control, we explored whether GABA or muscimol treatment of

**GABA accumulation in guard cells contributes to the regulation of transpiration and drought performance.** Stomatal control is explicitly linked with the regulation of plant water loss, which impacts the survival of plants under drought[7]; the wider the stomatal aperture, the greater the water loss of plants, the poorer the survival of plants under a limited water supply, as excessive water use by the plant diminishes the availability of stored soil water. The observation that the stress-induced metabolite GABA[13] reduces plant water loss and improves WUE (Fig. 2d; Supplementary Fig. 2c)—key factors underpinning drought tolerance[27]—implicates GABA as novel signal regulating plant drought resilience. Therefore, to examine the hypothesis that endogenous GABA concentration increases under a water deficit and acts as a signal, we first determined whether we could replicate the previously reported increases in GABA accumulation under drought[13,14,29] (Fig. 3; Supplementary Fig. 6). In wild-type plants, a drought treatment was applied by withholding watering, which resulted in the gradual depletion of soil gravimetric water and a reduction in leaf relative water content (RWC) (Supplementary Fig. 6a, b). We found that GABA accumulation in drought stressed leaves increased by 35% compared to that of well-watered leaves (water versus drought at

7 days: $1.07 \pm 0.08$ versus $1.44 \pm 0.11$ nmol mg$^{-1}$ FW) (Supplementary Fig. 6c).

To investigate whether GABA has a role during drought, we obtained *Arabidopsis* T-DNA insertional mutants for the major leaf GABA synthesis gene, *Glutamate Decarboxylase 2* (*GAD2*)[29]. Both *gad2-1* and *gad2-2* had >75% less GABA accumulation in leaves than in wild-type plants, whilst GABA concentrations in roots were unchanged (Fig. 3a; Supplementary Fig. 6d–f). Furthermore, leaves of *gad2* plants did not accumulate additional GABA under drought conditions unlike wild-type controls where GABA increased by 45% after 3 days, and was maintained at this elevated level after 7 days of drought (Fig. 3a). Under standard conditions, both *gad2* mutant lines exhibited greater stomatal conductance and wider stomatal pores than wild-type plants (Fig. 3b; Supplementary Fig. 6g), whereas stomatal density was identical to wild type (Supplementary Fig. 6h). The application of exogenous GABA to *gad2* leaves inhibited stomatal pore aperture changes in response to light treatments (Supplementary Fig. 6i, j), indicating that *gad2* stomata would be competent in a GABA response if sufficient GABA was present. Furthermore, the aperture of GABA pre-treated *gad2* stomata after a dark-to-light transition were statistically insignificant from non-GABA treated wild-type stomata (Supplementary Fig. 6j), which is consistent with GABA playing a role in modulating opening of wild-type stomata under non-stressed conditions. It has been shown previously that both *GAD2* transcription and GABA accumulation exhibit diurnal regulation; GABA usually peaks at the end of the dark cycle prior to stomatal opening and reaches a minimum when stomatal conductance is at its maximum near subjective mid-day[30]. However, during stress, both *GAD2* transcript abundance and GABA accumulation remain high[30]. This suggests GABA may further minimise stomatal opening under stress and contribute to drought tolerance.

Under drought, the leaf RWC of *gad2* plants lowered more quickly than in wild type (Fig. 3c). Transcriptional profiles of key ABA-marker gene (*RD22*) and GABA-related genes (other than *GAD2*) were similar in wild type and *gad2* lines, although *RD29A* was significantly higher in *gad2-1* than wild type and *gad2-2* on day 0 and day 7 of the drought treatment (Supplementary Fig. 7), which is consistent with the lower RWC of *gad2-1* after 7 days (Fig. 3c). These results confirm that *GAD2* is critical for leaf GABA production under stress, and suggests that GABA itself may regulate plant water loss and drought tolerance[29].

Histochemical staining corroborated that *GAD2* is highly expressed in leaves, particularly in guard cells[29] (Supplementary Fig. 8a, b). GAD2 is a cytosolic enzyme[31]; to examine if cytosolic GABA biosynthesis within the guard cell was sufficient to modulate transpiration we expressed—specifically in the guard cell[32]—a constitutively active form of *GAD2* (*GAD2Δ*) that has a C-terminal autoinhibitory domain removed[31,33] (Fig. 4a). This led to a large increase in leaf GABA accumulation (Fig. 4b) and to complementation of the steady-state stomatal conductance and aperture phenotypes of *gad2* plants to wild-type levels (Fig. 4c; Supplementary Fig. 8c, d). At the same time, no change in stomatal density or leaf ABA accumulation was detected under standard conditions (Supplementary Fig. 8e, f), suggesting the complementation of the *gad2* phenotype was due to the restoration of GABA synthesis in the guard cell. Other phenotypes restored to wild-type levels by guard cell-specific expression of *GAD2Δ* included the exaggerated stomatal opening and closure kinetics and decreased instantaneous iWUE/WUE of *gad2-1* (Fig. 4d–f; Supplementary Fig. 8g–i). The drought sensitivity of *gad2*, compared to wild type, was also abolished by guard cell-specific expression of *GAD2Δ* (Fig. 4g, h). This demonstrates GABA synthesis in guard cells was sufficient to modulate stomatal movement, regulate water loss and improve drought resilience.

To examine whether GABA metabolism can be modulated to improve drought resilience beyond wild-type levels, *GAD2Δ* was expressed specifically in the guard cells of wild-type *Arabidopsis* plants (Fig. 5a), this resulted in leaf GABA concentrations being increased to beyond wild-type levels (Fig. 5b). The steady-state stomatal conductance of the GABA overproducing transgenic plants in standard and drought conditions was lowered compared to wild-type plants (Fig. 5c). Consistent with this, the plants overexpressing *GAD2Δ* in the wild-type background maintained higher leaf RWC than wild-type plants after 10 days of drought treatment (Fig. 5d, e). Furthermore, a greater percentage of plants overexpressing *GAD2Δ* in the wild-type background survived following re-watering after a 12-day drought treatment (Supplementary Fig. 9). As such, we show here that GABA overproduction can reduce water loss and improve drought resilience.

**Guard cell cytosolic GABA modulates stomatal movement and drought resilience.** Our data show that although guard cell synthesised GABA can rescue the *gad2* phenotype, it is clear that exogenously applied GABA can also modulate stomatal movement (e.g. Fig. 2 for wild type or Supplementary Fig. 6i, j for *gad2*). It is known that GABA can pass the membrane through a variety of transporters[34–36], so it is unclear whether the site of guard cell GABA action is from the apoplast or cytoplasm. We expressed *GAD2Δ* specifically in the spongy mesophyll[37], adjacent to the abaxial stomatal layer, to test whether it could complement *gad2* (Supplementary Fig. 10a, b). This resulted in a significant increase in leaf GABA, but no change in stomatal conductance (Supplementary Fig. 10c, d). As such, unlike guard cell-specific expression, *GAD2Δ* in the spongy mesophyll was insufficient to complement the *gad2-1* phenotype.

To further probe the role of guard cell synthesised GABA, we expressed full-length *GAD2* under the guard cell-specific promoter (*gad2-1/GC1::GAD2*) (Supplementary Fig. 11a). This form of GAD2 requires activation by Ca$^{2+}$/calmodulin or low pH to synthesise GABA[14]. Interestingly, guard cell-specific expression of full-length *GAD2* failed to complement the high stomatal conductance of the *gad2-1* line to wild-type levels under standard conditions, whereas its constitutive expression (driven by pro35S-CAMV) did (Supplementary Fig. 11b, f, g). Under drought, the *gad2-1/GC1::GAD2* lines increased GABA production, reduced their stomatal conductance significantly more than that of *gad2-1* plants and had a comparable leaf RWC to wild-type plants following 5 days of drought (Supplementary Fig. 11c–e). This suggests that activation of full-length GAD2 via its regulatory domain[31] is important in stimulating GABA production under drought in guard cells.

We extended our investigation of GABA's site of action through an epidermal peel experiment. We compared the effects of exogenously applied muscimol or muscimol-BODIPY, a muscimol molecule conjugated with a BODIPY fluorophore, which is active against GABA targets in plants and animals, but lacks cell-membrane permeability[38,39]. We found that unlike muscimol, membrane impermeable muscimol-BODIPY was unable to inhibit stomatal opening or closure (Supplementary Fig. 12). This result—alongside the differential effects of *gad2* complementation by full-length *GAD2* when expressed constitutively or solely in the guard cell (Supplementary Fig. 11a–e) —provides further evidence that GABA is likely to pass the plasma membrane and that it acts from the cytosol, consistent with our feeding assays (e.g. Fig. 2c). Collectively, the data in this section demonstrate that guard cell-specific cytosolic GABA accumulation is sufficient and necessary for controlling stomatal aperture and transpiration under drought, but suggests a role for other cell types in fine-tuning GABA signals under standard conditions.

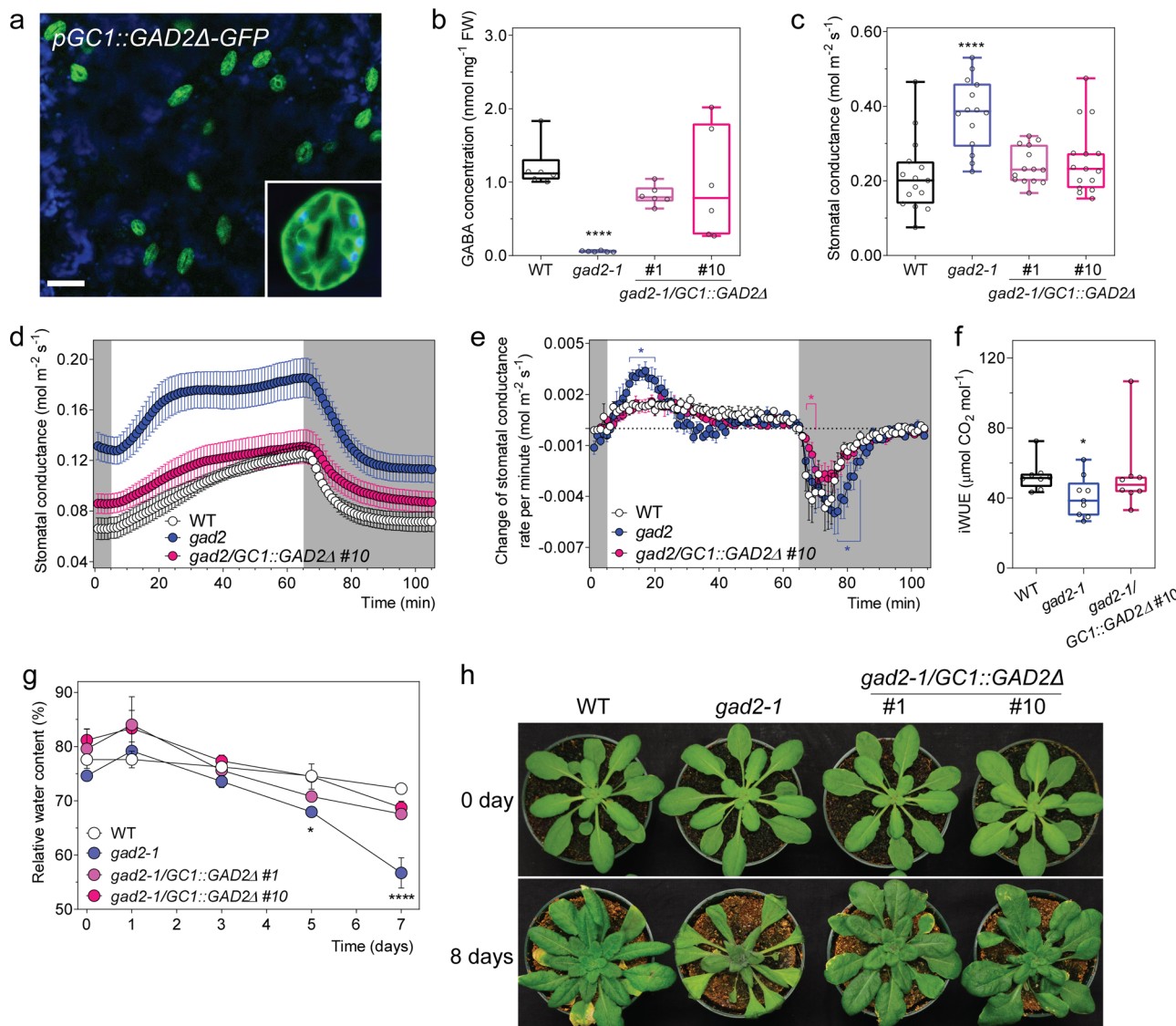

**Fig. 4 Guard cell GABA regulates water loss and drought tolerance. a** Representative confocal images of *gad2*-1 plants expressing *GC1::GAD2Δ-GFP* (*gad2-1/GC1::GAD2Δ-GFP*); GFP fluorescence and chlorophyll autofluorescence (blue) of the leaf abaxial side of 3–4-week-old *gad2-1/GC1::GAD2Δ-GFP* plant indicates that the *GC1* promoter drives *GAD2Δ* expression specifically in guard cells, similar pattern images are obtained from multiple *gad2-1/GC1::GAD2Δ-GFP* plants, scale bars = 50 μm. **b** Leaf GABA accumulation of 5–6-week-old *A. thaliana* WT, *gad2-1*, *gad2-1/GC1::GAD2Δ* #1 and #10 plants grown under control conditions, n = 6. **c** Stomatal conductance of WT (n = 15), *gad2-1* (n = 14), *gad2-1/GC1::GAD2Δ* #1 (n = 14) and #10 (n = 15) plants under control conditions determined using an AP4 porometer, data collected from two independent batches of plants. **d** Stomatal conductance of WT, *gad2-1* and *gad2-1/GC1::GAD2Δ* #10 plants in response to dark (shaded region) and 150 μmol m$^{-2}$ s$^{-1}$ light (white region), measured using a LI-COR LI-6400XT. **e** Change in stomatal conductance each minute calculated using dConductance/dt (min) of the data represented in **d**. **f** iWUE of WT, *gad2-1* and *gad2-1/GC1::GAD2Δ* plants was calculated based on the ratio of photosynthetic rate (Supplementary Fig. 8h) versus stomatal conductance represented in **d**; n = 8 individual plants for WT, n = 9 for *gad2-1* and n = 8 for *gad2-1/GC1::GAD2Δ* #10, data collected from two independent batches of plants (**d**–**f**). **g** Relative leaf water content of WT, *gad2-1*, *gad2-1/GC1::GAD2Δ* #1 and #10 plants following drought treatment for 0, 1, 3, 5 and 7 days; n = 4 for 0, 1, 3 and 5 days samples and n = 5 for 7 days samples, except that n = 3 for 0-day *gad2-1* and 1-day *gad2-1/GC1::GAD2Δ* #1. **h** Representative images of WT, *gad2-1*, *gad2-1/GC1::GAD2Δ* #1 and #10 plants (shown in **i**) before (0 day) and after (8 days) drought treatment as indicated. All data are plotted with box and whiskers plots: whiskers plot represents minimum and maximum values, and box plot represents second quartile, median and third quartile (**b**, **c**, **f**), or data are represented as mean ± s.e.m (**d**, **e**, **g**); statistical difference was determined using by two-sided Student's *t* test (**f**), one-way ANOVA (**b**, **c**) or two-way ANOVA (**e**, **g**); *P < 0.05 and ****P < 0.0001.

**GABA signalling regulating WUE and drought resilience is ALMT9 dependent**. ALMTs are plant-specific anion channels that share no homology to Cys-loop receptors except a region of 12 amino acid residues predicted to bind GABA in GABA_A receptors[14,18]. In animals, ionotropic GABA receptors are stimulated by GABA; in contrast, anion currents through ALMTs are inhibited by GABA[10,11]. There are a number of *ALMTs* expressed in guard cells that contain the putative GABA binding

motif and have the potential to transduce the GABA signal, with most having been shown to have a role in regulating stomatal movement[20–22,40]. For instance, ALMT12 (also called QUAC1, quickly-activation anion conductance 1) is a plasma membrane localised anion channel, which moves anions out of the guard cell during guard cell closure[20].

Under the conditions tested here, the impact of GABA on stomatal closure appears to be limited to epidermal peels, it is not

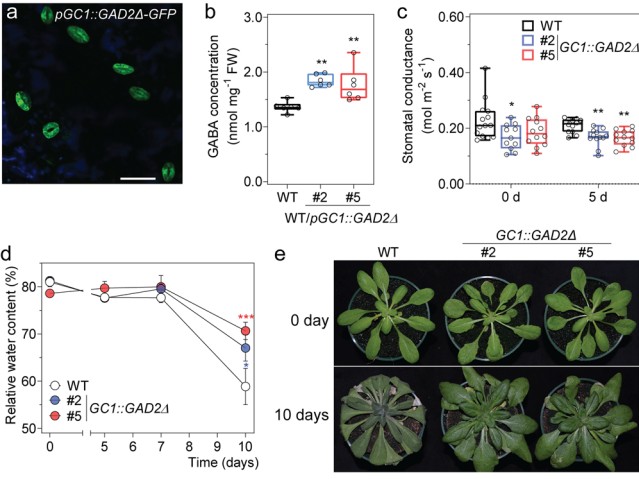

**Fig. 5 Guard cell overexpression of *GAD2Δ* decreases plant water loss and increases drought survival. a** Representative confocal images of *A. thaliana* wild-type plants expressing *GC1::GAD2Δ-GFP*; GFP fluorescence and chlorophyll autofluorescence (blue) of the leaf abaxial side of 3–4-week-old plants, similar pattern images are obtained from multiple wild-type plants expressing *GC1::GAD2Δ-GFP* plants, scale bars = 50 μm. **b** GABA accumulation in the leaves of 5–6-week-old Arabidopsis wild type, *GC1::GAD2Δ* #2 and #5 plants; *n* = 6. **c** Stomatal conductance of WT, wild-type Arabidopsis expressing *GAD2Δ* in the guard cells using the *GC1* promoter—*GC1::GAD2Δ* #2 and #5 plants before (0 day) and after (5 days) drought treatment determined using an AP4 porometer; *n* = 14 for WT, *n* = 11 for *GC1::GAD2Δ* #2 and *n* = 15 for *GC1::GAD2Δ* #2 at 0 day and *n* = 12 for WT, *GC1::GAD2Δ* #2 and #5 at 5 days. **d** Relative leaf water content of WT, *GC1::GAD2Δ* #2 and #5 plants following drought treatment for 0, 5, 7 and 10 days; *n* = 6 for 0 and 5 days all samples, except *n* = 18 for WT at 10 days, *n* = 12 for *GC1::GAD2Δ* #2 at 10 days and *n* = 13 for *GC1::GAD2Δ* #5 at 10 days. **e** Representative images of WT, *GC1::GAD2Δ* #2 and #5 plants before (0 day) and after (10 days) drought treatment as indicated. Pot size 2.5 inch diameter × 2.25 inch height (LI-COR). All data are plotted with box and whiskers plots: whiskers plot represents minimum and maximum values, and box plot represents second quartile, median and third quartile (**b**, **c**), or data are represented as mean ± s.e.m (**d**); statistical difference was determined using one-way ANOVA (**b**, **c**) or two-way ANOVA (**d**); *$P$ < 0.05, **$P$ < 0.01 and ***$P$ < 0.001.

seen in intact leaves. However, epidermal peels still represent an assay system that can be used to test whether ALMT might transduce the inhibitory effect of GABA on closure. We observed that, unlike wild-type plants, stomatal closure in *almt12* knockouts was insensitive to GABA or muscimol when transitioning from light-to-dark (Supplementary Fig. 13a). In contrast, stomatal opening of *almt12* lines showed wild-type-like sensitivity to GABA or muscimol when transitioning from dark to light (Supplementary Fig. 13b). These data indicate that ALMT12 is a plasma membrane GABA target that affects stomatal closure in response to dark—in epidermal peels at least.

However, if GABA inhibition of ALMT12/QUAC1 played a significant role during drought, then the resulting inhibition of closure would translate into an increase in water loss compared to wild-type plants during closure. As we found no evidence that GABA had an effect on closure in intact leaves, under a light-to-dark transition as measured by stomatal conductance or transpiration (Fig. 2c; Supplementary Fig. 2a), and the fact that GABA accumulation led to a net decrease in water loss and improvement in drought resilience, ALMT12 is unlikely to be a major target contributing to this outcome. We therefore focused on tonoplast-localised ALMTs that are involved in stomatal pore

opening[21], as this is the process where GABA has its predominant affect in intact leaves.

ALMT9 is the major tonoplast-localised channel involved in anion uptake into guard cell vacuoles during stomatal opening, but has no documented role in closure[21]. We hypothesised that GABA might target and inhibit ALMT9 activity to reduce the rate or extent of stomatal opening. We initially attempted in vitro electrophysiological studies to examine the impact of GABA on ALMT9-induced currents, but were unable to consistently detect stable currents following heterologous expression in either *Xenopus laevis* oocytes or tobacco mesophyll cells[21,41]. Therefore, we examined the potential regulation of ALMT9 by GABA by focusing solely on in planta studies as it is difficult to faithfully replicate regulatory pathways from guard cells in heterologous systems, e.g.[42–47]. In the first instance, we independently crossed two *almt9* alleles (*almt9-1* and *almt9-2*) with *gad2-1*. We found that, similar to *gad2*, both double mutants (*gad2-1/almt9-1* and *gad2-1/almt9-2*) maintained low GABA accumulation in their leaves (Fig. 6a, b; Supplementary Fig. 14a, e). However, both *gad2-1/almt9-1* and *gad2-1/almt9-2* had wild-type-like stomatal conductance and aperture unlike *gad2-1* where both these parameters are high (Fig. 6c, d; Supplementary Fig. 14d, f). Furthermore, guard cell-specific complementation of *gad2-1/almt9-1* by *GAD2Δ* did not alter stomatal conductance (Supplementary Fig. 14a–d). Collectively, these data are consistent with ALMT9 being required for GABA to regulate gas exchange via stomatal control. An interesting additional observation was that the loss of *ALMT9* in *gad2-1* also resulted in ABA inducing stomatal pore closure to wild-type levels (Supplementary Fig. 14g–j), indicating that, although ALMT9 is a channel that regulates stomatal opening, it can influence the extent to which stomatal pores close under certain conditions (in epidermal peels at least). The incomplete stomatal closure of *gad2* coupled to its greater stomatal opening may further contribute to its drought sensitivity. These findings are consistent with the regulation of stomatal aperture being a dynamic equilibrium between the pathways that regulate stomatal opening and closure, with stomatal aperture being weighted towards a particular state dependent upon the dominant stimuli[48,49].

To further test whether ALMT9 transduces GABA signalling, we examined the effect of GABA on regulating stomatal opening in *almt9* mutant plants. In wild-type plants, we previously showed that light-induced stomatal opening was inhibited by exogenous GABA (Fig. 2a) or muscimol (Supplementary Fig. 1a). In *almt9* lines, exogenous GABA or muscimol did not antagonise stomatal opening (Fig. 7a, b; Supplementary Fig. 15a, b), whereas dark-induced stomatal closure in *almt9* retained its GABA sensitivity (Fig. 7c, d; Supplementary Fig. 15c, d). These results are consistent with GABA reducing stomatal opening via negative regulation ALMT9-mediated Cl⁻ uptake into guard cell vacuoles. Furthermore, it strongly indicates the corollary of this finding, that the higher stomatal conductance phenotype of *gad2* is the result of greater ALMT9 activity due to its lack of inhibition by GABA.

We tested this hypothesis by attempting to complement *almt9* plants with either the native channel or a site-directed ALMT9 mutant (ALMT9$^{F243C/Y245C}$). The mutations within ALMT9$^{F243C/Y245C}$ are in the 12 amino acid residue motif that shares homology with a GABA binding region in mammalian GABA$_A$ receptors[14,18]. Mutations in the aromatic amino acid residues in this motif have been shown for other ALMTs to result in active channels that are not inhibited by GABA when tested in heterologous systems[36,39] (Fig. 8; Fig. 9). However, no in planta tests have been conducted to date—for any ALMT—to determine whether mutations in this region result in a transport competent protein that lacks GABA sensitivity. Here, we observed that *ALMT9* and *ALMT9*$^{F243C/Y245C}$ had similar

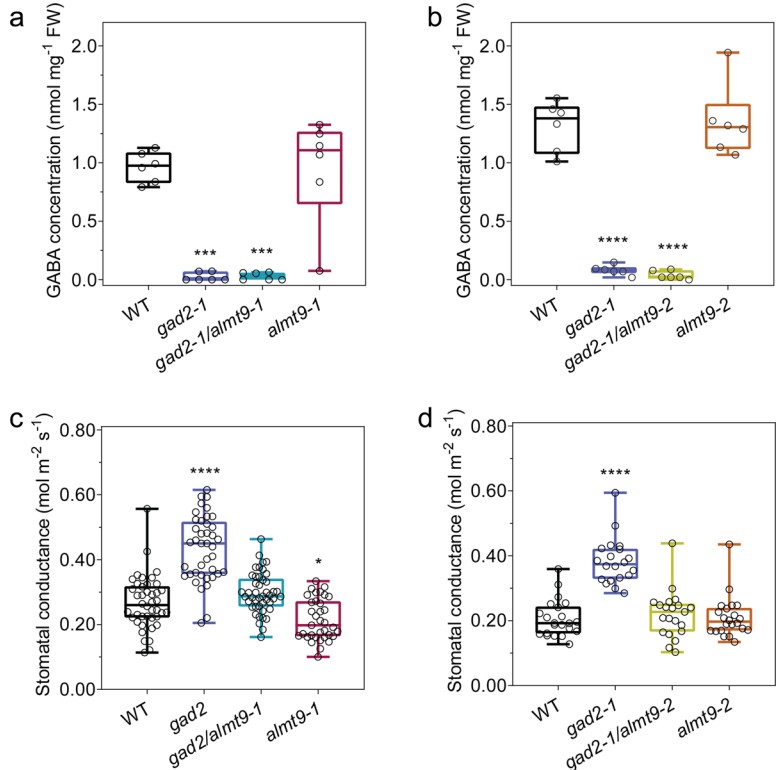

**Fig. 6 The loss of ALMT9 suppresses the *gad2* mutant stomatal phenotype. a–d** Leaf GABA concentration (**a**, **b**) and stomatal conductance (**c**, **d**) of 5–6-week-old *A. thaliana* WT, *gad2-1*, *gad2-1/almt9-1*, *almt9-1*, *gad2-1/almt9-2* and *almt9-2* plants; $n = 6$ plants (**a**, **b**); $n = 42$ for WT, $n = 40$ for *gad2-1*, $n = 45$ for *gad2-1/almt9-1* and $n = 35$ for *almt9-1*, data collected from four independent batches of plants (**c**); $n = 22$ for WT, $n = 20$ for *gad2-1*, $n = 21$ for *gad2-1/almt9-2* and $n = 22$ for *almt9-2*, data collected from two independent batches of plants (**d**); data (**a**, **c**) were extracted respectively from Supplementary Fig. 13b, c. All data are plotted with box and whiskers plots: whiskers plot represents minimum and maximum values, and box plot represents second quartile, median and third quartile; statistical difference was determined by one-way ANOVA, *$P < 0.05$, ***$P < 0.001$ and ****$P < 0.0001$.

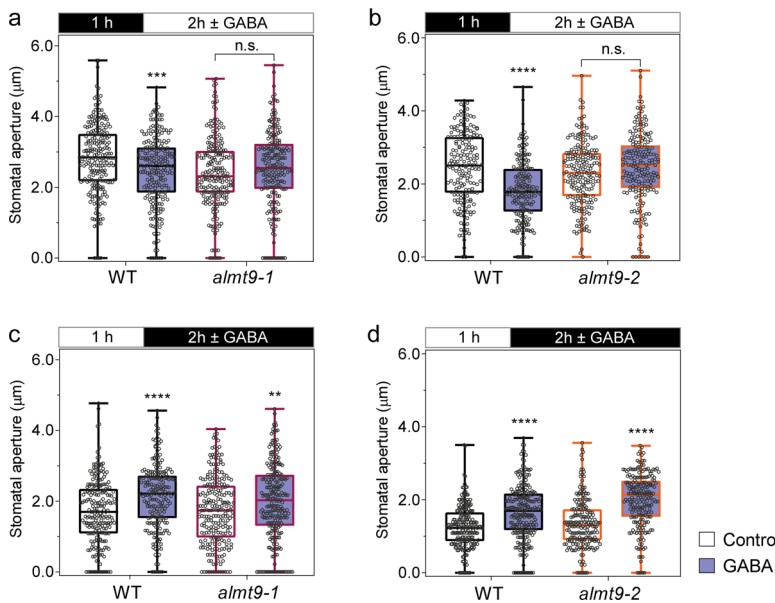

**Fig. 7 The loss of *ALMT9* abolishes GABA inhibition of stomatal opening but does not affect closure. a–d** *Arabidopsis* WT and *almt9* knockout plant stomatal aperture in response to light or dark. Epidermal strips were pre-incubated in stomatal measurement buffer for 1 h under dark (**a**, **b**) or light (**c**, **d**), followed by 2 h in light (**a**, **b**) or dark (**c**, **d**) as indicated by black (dark) or white (light) bars above graphs with ±2 mM GABA; $n = 236$ for WT and $n = 221$ for *almt9-1* with control treatment, $n = 229$ for WT and $n = 215$ for *almt9-1* with GABA treatment (**a**); $n = 223$ for WT and $n = 242$ for *almt9-1* with control treatment, $n = 215$ for WT and $n = 256$ for *almt9-1* with GABA treatment (**b**); $n = 183$ for WT and $n = 189$ for *almt9-2* with control treatment, $n = 210$ for WT and $n = 197$ for *almt9-2* with GABA treatment (**c**); $n = 236$ for WT and $n = 243$ for *almt9-2* with control treatment, $n = 202$ for WT and $n = 220$ for *almt9-2* with GABA treatment (**d**). All data are plotted with box and whiskers plots: whiskers plot represents minimum and maximum values, and box plot represents second quartile, median and third quartile; statistical difference was determined by two-way ANOVA, **$P < 0.01$, ***$P < 0.001$ and ****$P < 0.0001$.

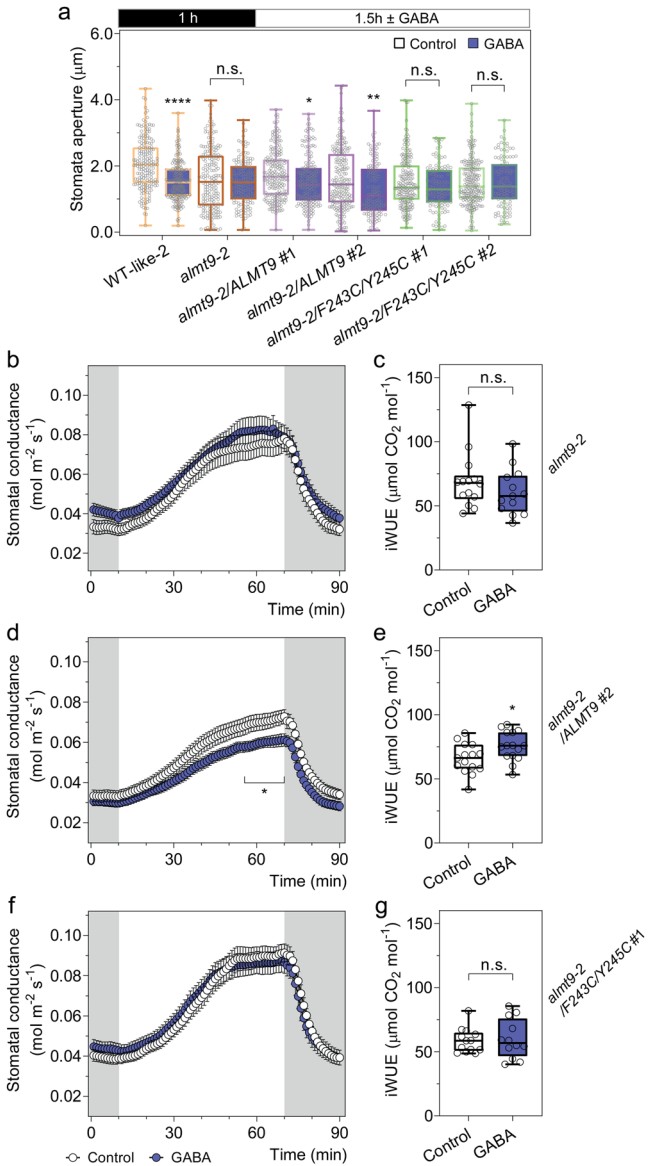

**Fig. 8 *ALMT9* but not *ALMT9*[F243C/Y245C] restores the GABA sensitivity of *almt9-2*. a** Stomatal aperture measurement of *A. thaliana* WT, *almt9-2* and complementation lines. Epidermal strips were pre-incubated in stomatal measurement buffer for 1 h under dark, followed by a 1.5 h dark-to-light transition, as indicated above graphs by black (dark) or white (light) bars, ±2 mM GABA; $n = 189$ (control) and $n = 195$ (GABA) for WT-like 2 (segregated from *almt9-2*)[21], $n = 197$ (control) and $n = 153$ (GABA) for *almt9-2*, $n = 213$ (control) and $n = 178$ (GABA) for *almt9-2* complement with *35S::ALMT9* #1 (*almt9-2/ALMT9* #1), $n = 219$ (control) and $n = 127$ (GABA) for *almt9-2/ALMT9* #2, $n = 195$ (control) and $n = 115$ (GABA) for *almt9-2* complemented with *35S::ALMT9* with double mutation F243C/Y245C (*ALMT9*[F243C/Y245C]) targeting the putative GABA interaction residues[18,36,39] (*almt9-2/F243C/Y245C* #1), $n = 221$ (control) and $n = 109$ (GABA) for *almt9-2/F243C/Y245C* #2 with control treatment. **b–g** Leaf feeding assay of *almt9-2* and complementation lines. Stomatal conductance of detached leaves from 5–6-week-old Arabidopsis *almt9-2*, *almt9-2/ALMT9* #2 and *almt9-2/F243C/Y245C* #1 plants was recorded using a LI-COR LI-6400XT in response to dark (shaded region) and 200 µmol m$^{-2}$ s$^{-1}$ light (white region), fed with artificial xylem sap solutions ± 4 mM GABA (**b, d, f**). The iWUE of *almt9-2* (**c**), *almt9-2/ALMT9* #2 (**e**) and *almt9-2/F243C/Y245C* #1 (**g**) detached leaves was calculated based on the ratio of photosynthetic rate (Supplementary Fig. 17b, e, h) versus stomatal conductance (**b, d, f**); $n = 14$ (control) and $n = 13$ (GABA) for *almt9-2* (**b, c**); $n = 15$ (control and GABA) for *almt9-2/ALMT9* #2 (**d, e**); $n = 13$ (control) and $n = 12$ (GABA) for *almt9-2/F243C/Y245C* #1 (**f, g**). All data are plotted with box and whiskers plots: whiskers plot represents minimum and maximum values, and box plot represents second quartile, median and third quartile (**a, c, e, g**), or data are represented as mean ± s.e.m (**b, d, f**); statistical difference was determined by two-sided Student's *t* test; *$P < 0.05$, **$P < 0.01$, ****$P < 0.0001$ (**a–g**).

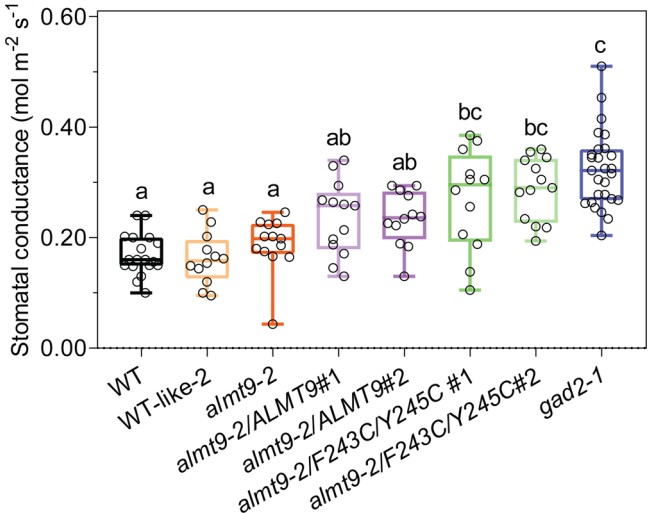

**Fig. 9 *ALMT9*[F243C/Y245C] increases steady-state stomatal conductance.** Stomatal conductance of 5–6-week-old Arabidopsis WT, *gad2-1*, *almt9-2* and complementation lines determined using an AP4 Porometer; $n = 18$ for WT, $n = 12$ for WT-like 2, *almt9-2/ALMT9* #2 and *almt9-2/F243C/Y245C* #1, $n = 13$ for *almt9-2*, *almt9-2/ALMT9* #1 and *almt9-2/F243C/Y245C* #2, $n = 27$ for *gad2-1*. All data are plotted with box and whiskers plots: whiskers plot represents minimum and maximum values, and box plot represents second quartile, median and third quartile; statistical difference was determined by one-way ANOVA, the letters a, b and c represent data groups that are not statistically different, $P < 0.05$.

expression in *almt9-2* complementation lines and the mutations (in ALMT9[F243C/Y245C]) did not alter the membrane localisation with both versions of the ALMT9 protein being clearly present on the tonoplast (Supplementary Fig. 16). Further, we found that similar to *almt9* lines, *almt9-2* expressing *ALMT9*[F243C/Y245C] was insensitive to GABA during a dark-to-light transition assayed on epidermal peels and detached leaves, for stomatal opening and stomatal conductance, respectively; this contrasts the GABA sensitivity of wild-type plants and plants expressing native *ALMT9* in the *almt9-2* background (Figs. 2c and 8a, b, d, f; Supplementary Fig. 17a, b, d, e, g, h). Furthermore, instantaneous iWUE/WUE of *almt9-2* was improved by native *ALMT9* complementation, but not *ALMT9*[F243C/Y245C] (Fig. 8c, e, g; Supplementary Fig. 17c, f, i). Steady-state stomatal conductance and aperture of *ALMT9*[F243C/Y245C] lines were also significantly greater than that of wild-type and *almt9* lines and were insignificant from *gad2-1* under standard conditions (Fig. 9; Supplementary Fig. 17j). This result indicates that we successfully complemented *almt9* with an active, but GABA-insensitive form of ALMT9, and that this increased transpirational water loss over wild-

type levels. These data are completely consistent with ALMT9 being a GABA target that regulates plant water loss, even under non-stressed conditions, through modulation of ALMT9 activity. The GABA effect is then amplified under a water deficit when GABA concentration increases. We propose that GABA accumulation has a role in promoting drought resilience by reducing the amplitude of stomatal re-opening each morning, which minimises whole plant water loss. As such, the GABA–ALMT pathway is a strong candidate for constituting the ABA-independent stress memory of a decreased soil water status that has been previously proposed without mechanistic attribution[50,51].

## Discussion

The data in this manuscript have unveiled a GABA signalling pathway in plants, which can be summarised by the simplified models presented in Fig. 10. We propose that cytosolic GABA signals, generated by GAD2, modulate stomatal opening, WUE and drought resilience transduced through negative regulation of ALMT9 activity (Fig. 10).

Collectively our use of leaf feeding, knockouts, complementation and point mutagenesis strongly suggests ALMT9 is an essential and major component transducing GABA signalling in guard cells during well-watered and drought conditions. As has become evident for other guard cell based signalling pathways through their examination over time[42–47], we are cognizant of the potential that other GABA response elements, including other ALMT, may be involved in transducing and fine-tuning this signalling pathway. Our finding that GABA does not impact stomatal closure in epidermal peels of *almt12* knockouts infers a potential role for this plasma membrane localised ALMT12 in transducing guard cell GABA signals. The fact that light-induced stomatal opening and dark-induced stomatal closure was completely GABA insensitive in *almt9xalmt12* knockouts (Supplementary Fig. 18) suggests that both channels have the potential to transduce the major effects of GABA in guard cells.

However, it is interesting that GABA inhibition of stomatal opening was consistently seen between epidermal peel assays and leaf feeding, whilst GABA only inhibited stomatal closure during isolated epidermal peel experiments, but not when it was fed to leaves. This suggests that GABA acts through ALMT12 on processes associated with stomatal closure, but in the context of an intact leaf this phenotype is lost, which is likely due to the loss of functional epidermal and/or mesophyll cells. This is consistent with the growing body of evidence that indicates stomatal aperture experiments on isolated epidermal peels require validation via studies on intact leaves to avoid overinterpreting potential artifacts from this reductionist system. However, it also means we cannot fully rule out whether GABA inhibition of stomatal closure does have a role under certain physiological scenarios that are yet to be identified. Therefore, in future studies, it would be pertinent to examine whether ALMT12-dependent GABA inhibition of stomatal closure has a physiological role in transducing GABA signals in conditions not examined here, and, more broadly, whether other ALMTs or additional elements are involved in GABA signal transduction.

ALMT activity appears to be regulated by a suite of factors including anions, ($Al^{3+}$ for ALMT1), pH, ATP, voltage and GABA[52]. As such, it is becoming clear that ALMTs have the potential to act as a key signalling hub in a variety of physiological processes. Following on from this study, leading on from the observed GABA modulation of ABA, $H_2O_2$ and coronatine effects on stomata, the investigation into cross-talk between GABA and other signals for ALMT9, in particular, and ALMTs, in general, provides the basis for future research areas. Such studies will be able to resolve questions such as 'whether GABA can act directly on guard cell ALMTs?', as appears to occur for wheat ALMT1[18,39], or 'whether other signalling intermediates are also involved?'. GABA inhibition of the wheat ALMT1 anion conductance was recently found to occur from the cytosol only, by reducing the open probability of the channel to anions[39]. However, that study was unable to determine whether this occurred through permeation of uncharged GABA through the ALMT

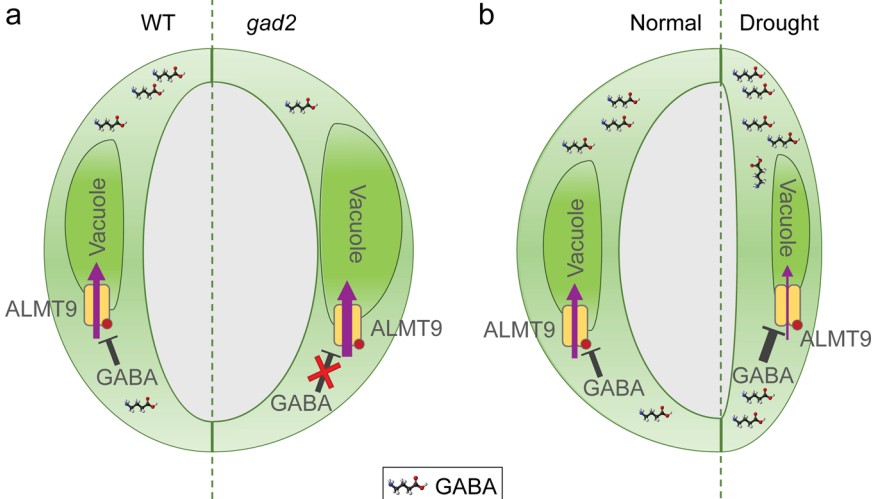

**Fig. 10 Proposed model of GABA-mediated signalling for the regulation of water use efficiency. a** Cytosolic guard cell GABA negatively regulates ALMT9-mediated anion uptake into guard cell vacuoles, which fine tunes stomatal opening (left guard cell of pair). Depletion of GABA accumulation in the leaves of *GAD2* loss-of-function mutant (*gad2*) de-regulates ALMT9, maximizing anion uptake and accumulation in guard cell vacuoles. This leads to a more open stomatal pore, greater water loss and lower WUE of plants (right guard cell of pair). This stomatal phenotype can be replicated by replacing F243/Y245 (red dot) with two cysteines, which abolishes GABA sensitivity of ALMT9. **b** Leaf GABA synthesized and accumulated during water deficit reduces ALMT9-mediated vacuolar anion uptake into guard cells, which requires amino acid residues F243/Y245 (red dot) (right guard cell of pair). This reduces stomatal opening, reducing the pore aperture and enhances plant WUE under drought stress compared to guard cells under standard conditions in the light (left guard cell of pair). Note: We have excluded ALMT12 from this model as we did not find a role for this protein in GABA modulation of water use efficiency in planta, despite its role in GABA modulation of stomatal aperture found within epidermal peels.

pore or through GABA binding modifying channel structure[39]. Cytosolic GABA inhibition was dependent upon the putative GABA binding residue F213 (equivalent to F243 in ALMT9, which is also predicted to face the cytosol)[39,53]. Our study therefore highlights the real need to definitively determine whether GABA binds to ALMTs or whether the identified amino acid residues affect GABA sensitivity independent of anion permeability through other means. For instance, future studies should address whether GABA permeability of ALMTs has a role in signal transduction in guard cells and the regulation of other physiological processes[36]. These later questions would be aided by the determination of GABA concentrations in different cell types and compartments to further understand the co-ordination of GABA signalling across membranes, leaves and other organs, and this could be achieved through the deployment of novel GABA sensors, as recently used in animal tissues[54,55].

GABA concentration oscillates over diel cycles and increases in response to multiple abiotic and biotic stresses including drought, heat, cold, anoxia, wounding pathogen infection and salinity[13]. ALMTs have been implicated in modulating multiple developmental and physiological processes in plants[20–22,56–58] including those underpinning nutrient uptake and fertilization that are affected by GABA[18,59,60]. Therefore, the discovery that GABA regulates ALMT to form a physiologically relevant signalling mechanism in guard cells is likely to have broad significance beyond stomata, particularly during plant responses to environmental transitions and stress.

GABA's effect on stomata appears to be conserved across a large range of crops from diverse clades including important monocot and dicot crops (Supplementary Fig. 5), indicating that GABA may well be a stomatal signal of economic significance. As we find that the genetic manipulation of cell-type specific GABA metabolism can reduce water loss leading to improved drought performance, our work opens up alternative ways for manipulating crop stress resilience. This statement is tempered in the knowledge that GABA modulated stomatal signalling in the face of another signal and did not stimulate changes in stomatal aperture itself. GABA's role appears to be that of fine-tuning stomatal aperture. Our data suggest that GABA modulated stomatal movement occurs in response to light and dark and low concentrations of signal intermediates, but in the face of a strong stress stimulus its affects may be overridden. As such, GABA may well provide a direct link between the metabolic status of the cell —GABA being produced in the cytosol in times of stress as a bypass of several reactions of the TCA cycle—to regulate and sustain a certain physiological process prior to it being shut down via a more severe stress response pathway. More broadly, this study also provides proof that GABA is a plant signalling molecule and not just a plant metabolite[12,16], and in so doing, we conclude that GABA is an endogenous signalling molecule beyond the animal and bacterial kingdoms, enacted through distinct and organism specific mechanisms.

## Methods

**Plant materials and growth conditions**. All experiments were performed on *A. thaliana* were in the Columbia-0 (Col-0) ecotype background, unless stated. *Arabidopsis* wild type, T-DNA insertion mutant and other transgenic plants were germinated and grown on ½ Murashige and Skoog (MS) medium with 0.8% phytagel for 10 days before being transferred to soil for growth in short-day conditions (100–120 μmol m$^{-2}$ s$^{-1}$, 10 h light/14 h dark) at 22 °C. The T-DNA insertion mutant *gad2-1* (GABI_474_E05) and *gad2-2* (SALK_028819) were obtained from the Arabidopsis Biological Resource Centre (ABRC). *gad2-1* was selected using primer sets:

gad2_LP1 (5′-TATCACGCTAACACCTAACGC-3′), gad2_RP1 (5′-TTCAAGGTTTGTCGGTATTGG-3′) and GABI_LB (5′-GGGCTACAC TGAATTGGTAGCTC-3′) for removing the second T-DNA insert; gad2_LP2 (5′-ACGTGATGGATCCAGACAAAG-3′), gad2_RP2 (5′-TCTTCATTTCCAC ACAAAGGC-3′) and GABI_LB for isolation of the *GAD2* (At1g65960) T-DNA

insertion. *gad2-2* was selected using primer sets: gad2-2_LP (5′-AGTTGTATGAA AGTTCATGTGGC-3′), gad2-2_RP (5′-TCGACCACGAGATTTTAATGG -3′) and SALK_LB (5′-ATTTTGCCGATTTCGGAAC-3′). *almt9-1* (SALK_055490), *almt9-2* (WiscDsLox499H09)), *almt12-1* (SM_3_38592) and *almt12-2* (SM_3_1713) were selected as described previously[20,21]. The double mutant lines *gad2/almt9-1*, *gad2-1/almt9-2*, *almt9-2/12-1* and *almt9-2/12-2* were obtained, respectively, from crossing the respective mutants. The mesophyll enhancer-trap line JR11-2 in the Col-0 background was kindly provided by K. Baerenfaller (ETH Zurich)[61]. JR11-2 (Col-0) and *gad2-1/*JR11-2 were segregated from crossing *gad2-1* with JR11-2. JR11-2 was selected using primer sets: JR11-2_LP (5′-TTATTAGGG AAATTACAAGTTGC-3′), JR11-2_RP (5′-AGACACATTTAATAACATTACAAC AAA-3′) and JR11-2_LB (5′-GTTGTCTAAGCGTCAATTTGTTT-3′)[62]. All experiments were performed on stable T$_3$ transgenic plants or confirmed homozygous mutant lines. The other plants *V. faba*, *N. benthamiana* and *G. max* were grown in soil in long-day conditions (400 μmol m$^{-2}$ s$^{-1}$, 16 h light/8 h dark, 28 °C/25 °C). *H. vulgare* (barley) cv. Barke was grown in a hydroponic system with half-strength Hoagland's solution in long-day conditions (150 μmol m$^{-2}$ s$^{-1}$, 16 h light/8 h dark, 23 °C)[63].

**Gene cloning and plasmid construction**. For guard cell-specific complementation, the constitutively active form of *GAD2* with a truncation of the calmodulin binding domain (*GAD2Δ*)[31,33] and the full-length *GAD2* coding sequence (*GAD2*) was driven by a guard cell-specific promoter *GC1* (−1140/+23)[32], as designated *GC1::GAD2Δ* and *GC1::GAD2*, respectively. PCR reactions first amplified the truncated *GAD2* with a stop codon and *GC1* promoter (*GC1*) separately using Phusion® High-Fidelity DNA Polymerase (New England Biolabs) with the primer sets: GAD2_forward (5′-CACTACTCAAGAAATATGGTTTTGACAAAAACC GC-3′) and GAD2_truncated_reverse (5′-TTATACATTTTCCGCGATCCC-3′); GC1_forward (5′-CACCATGGTTGCAACAGAGAGGATG-3′) and GC1_reverse (5′-ATTTCTTGAGTAGTGATTTTGAAG-3′). This was followed by an overlap PCR to fuse the *GC1* promoter to *GAD2Δ* (*GC1::GAD2Δ*) with the GC1_forward and GAD2_truncated_reverse primer set. The same strategy was used to amplify *GC1::GAD2Δ* without a stop codon (*GC1::GAD2Δ-stop*), *GC1::GAD2* and *GC1:: GAD2* without a stop codon (*GC1::GAD2-stop*) with different primer sets: (1) *GC1:: GAD2Δ-stop* amplified with GAD2_forward and GAD2_truncated_reverse (5′-TACATTTTCCGCGATCCCT-3′); (2) *GC1::GAD2* amplified with GAD2_forward and GAD2_reverse (5′-TTAGCACACACCATTCATCTTCTT-3′) and (3) *GC1::GAD2-stop* amplified with GAD2_forward and GAD2-stop_reverse (5′-CACACCATTCATCTTCTTCC-3′). The fused PCR products were cloned into the pENTR/D-TOPO vector via directional cloning (Invitrogen). pENTR/D-TOPO vectors containing *GC1::GAD2Δ* or *GC1::GAD2* were recombined into a binary vector pMDC99[64] by an LR reaction using LR Clonase II Enzyme mix (Invitrogen) for guard cell-specific complementation, after an insertion of a NOS Terminator into this vector. A pMDC99 vector was cut by *Pac*I (New England Biolabs) and ligated with NOS terminator flanked with *Pac*I site using T4 DNA ligase (New England Biolabs). This NOS terminator flanked with *Pac*I site was amplified with primer set: nos_PacI_forward (5′-TACGTTAATTAAGAATTTCCCCGAT-3′) and nos_PacI_reverse (5′-GCATTTAATTAAAGTAACATAGATGACACC-3′) and cut by restriction enzyme *Pac*I before T4 DNA ligation. *GC1::GAD2Δ-stop* and *GC1::GAD2-stop* were recombined from the pENTR/D-TOPO vector into a pMDC107 vector that contained a GFP tag on the C-terminus (*GC1::GAD2Δ-GFP* and *GC1::GAD2-GFP*)[64].

To create *GAD2* complementation driven by a constitutive 35S promoter, the full-length *GAD2* was also amplified using primer set GAD2_forward2 (5′-CACC ATGGTTTTGACAAAAACCGC-3′) and GAD2_reverse and cloned into pENTR/ D-TOPO vector via directional cloning (Invitrogen), followed by an LR reaction recombinant into pMDC32[64]. For mesophyll specific complementation, *GAD2Δ* with a stop codon was amplified with the GAD2_forward2 and GAD2_truncated_reverse primer set, and cloned into the pENTR/D-TOPO vector, followed by an LR reaction recombined into the pTOOL5 vector (*UAS::GAD2Δ*)[65].

For *almt9-2* complementation, the pART27 binary vector containing the *ALMT9* coding sequence[21] was used for native *ALMT9* complementation driven by the 35S promoter, and also used as a template for a site-direct mutagenesis PCR to replace F243 and Y245 of *ALMT9* with two cysteines (*ALMT9*$^{F243C/Y245C}$) using the primer sets: ALMT9_DoubleF (5′-GTTTAGGTGTTAATATGTGTATCTGT CCTATATGGGCTGGAGAGG-3′) and ALMT9_DoubleR (5′-CCATATAGGACA GATACACATATTAACACCTAAACTAACACCAGCACC-3′).

For *GAD2* expression analysis, a 1 kb sequence upstream of the *GAD2* start codon was designated as the *GAD2* promoter (*pGAD2*) and amplified using primer set proGAD2_F (5′-ATTTTGAATTTGCGGAGAATCT-3′) and proGAD2_R (5′-CTTTGTTTCTGTTTAGTGAAAGAGAA-3′). The *pGAD2* PCR product was cloned into pCR8/GW/TOPO via TA cloning and recombined via an LR reaction into the pMDC162 vector containing the *GUS* reporter gene for histochemical assays[64]. The binary vectors, pMDC32, pMDC99, pMDC107, pMDC162, pTOOL5 and pART27 carrying sequence-verified constructs, were transformed into *Agrobacterium* strain AGL1 for stable transformation in *Arabidopsis* plants.

**Stomatal aperture and density measurement**. Soil-grown *Arabidopsis* (5–6-week-old) were used for stomatal aperture and density measurements. Two-to-three-week-old soybean, broad beans and barley and 5–6-week-old tobacco were used for

stomatal aperture assays. Epidermal strips from *Arabidopsis*, soybean, faba bean and tobacco were peeled from abaxial sides of leaves, pre-incubated in stomatal pore measurement buffer containing 10 mM KCl, 5 mM L-malic acid, 10 mM 2-ethanesulfonic acid (MES) with pH 6.0 by 2-amino-2-(hydroxymethyl)-1,3-propanediol (Tris) under light (200 μmol m$^{-2}$ s$^{-1}$) or darkness and transferred into stomatal pore measurement buffer with blind treatments as stated in the figure legend. For barley epidermal stomatal assays, a modified method was used[66]: the second fully expanded leaf from 2-week-old seedlings was used as experimental material, leaf samples were first detached and bathed in a modified measurement buffer (50 mM KCl, 10 mM MES with pH 6.1 by KOH) under light (150 μmol m$^{-2}$ s$^{-1}$) for 1.5 h or darkness for 1 h, then pre-treated in the same buffer with or without 1 mM GABA for 0.5 h; after this pre-treatment, samples were incubated in continuous dark, light, light-to-dark or dark-to-light transition for an additional 1 h as indicated in the figure legend before leaf epidermal strips were peeled for imaging. For *Arabidopsis* stomatal density measurement, epidermal strips were peeled from abaxial sides of young and mature leaves, three leaves per plants, three plants per genotype. Epidermal strips for both aperture and density measurement were imaged using an Axiophot Pol Photomicroscope (Carl Zeiss) apart from the barley epidermal strips imaged using an Nikon Diaphot 200 Inverted Phase Contrast Microscope (Nikon). Stomatal aperture and density were analyzed using particle analysis (http://rsbweb.nih.gov/ij/).

**Stomatal conductance measurement.** All stomatal conductance measurements were performed on 5–6-week-old *Arabidopsis* plants. The stomatal conductance determined by the AP4 Porometer (Delta-T Devices) was calculated based on the mean value from 2–3 leaf recordings per plant (Figs. 3b, 4c, 5c, 6c, d and 9; Supplementary Fig. 10d, 11d, g and 14c). The time-dependent stomatal conductance, transpiration and photosynthetic rate was recorded using LI-6400XT Portable Photosynthesis System (LI-COR Biosciences) equipped with an *Arabidopsis* leaf chamber fluorometer (under 150 μmol m$^{-2}$ s$^{-1}$ light with 10% blue light, 150 mmol s$^{-1}$ flow rate, 400 ppm CO$_2$ mixer, ~50 % relative humidity at 22 °C) as indicated (Fig. 4d; Supplementary Fig. 8g, h).

**ABA measurement.** The analysis of *Arabidopsis* leaf ABA concentration followed a method as described previously[67]. Briefly, >50 mg of ground fresh leaf samples were used to determine ABA concentration using an Agilent 6410 Series Triple Quad liquid chromatography (LC)-mass spectrometer (MS)/MS, equipped with Agilent 1200 series HPLC (Agilent Technologies) using a Phenomenex C18 column (75 mm × 4.5 mm × 5 μm) with a column temperature set at 40 °C. Solvents were nanopure water and acetonitrile, both with 0.05% acetic acid. Samples were eluted with a linear 15-min gradient from 10 to 90% acetonitrile. Compounds were identified by retention times and mass/charge ratio.

**Water-deficit drought assay.** Plants were germinated on ½ MS medium with 0.8% phytagel for 10 days in short-day conditions (100–120 μmol m$^{-2}$ s$^{-1}$, 10 h light/14 h dark) at 22 °C before being transferred to pots (size 2.5 inch diameter × 2.25 inch height, LI-COR Bioscience) with soil, containing coco peat/Irish peat (1:1 ratio). Prior to 10-day-old seedling transfer, all pots were weighed on an Ohaus ARA520 Adventurer Balance and soil was aliquoted into the pots within ±0.1 g between all replicates within an experimental run, randomly placed in growth cabinet and moved every other day in the same environmental conditions stated above. The starting weight varied amongst experimental runs dependent upon soil moisture (from 75 to 78 g). The drought assay was performed on 5–6-week-old *Arabidopsis* plants (Figs. 3, 4g, h and 5; Supplementary Figs. 6a–c, 9 and 11c–e). All plants were well-watered (saturated) the night before the drought assay, but not watered again during the assay. During the drought assay, all plants were randomly moved around once a day to avoid any bias of uneven light distribution or air flow within the cabinet that may differentially affect water loss.

At each sampling point, fresh weight of 2–3 leaves per plant was determined on an Ohaus Explorer E02140 balance (in Fig. 3a, Supplementary Figs. 6c and 11c, this occurred immediately after the rest of the leaf rosette was snap frozen in liquid nitrogen for later GABA measurement). Sampled leaves were then rehydrated to full turgid weight in ultrapure water overnight and measured after surface water was dried with paper towel. Dry weight was determined at 65 °C for 1 day. Leaf RWC was calculated as (Figs. 3c, 4g and 5d; Supplementary Figs. 6a and 11e)

$$RWC = \frac{Fresh\ weight - Dry\ weight}{Turgid\ weight - Dry\ weight} \times 100\% \quad (1)$$

At each sampling point, fresh soil weight of the whole pot (Mwet) and dry soil weight after drying the soil (Mdry) at 105 °C for 3 days was measured using an Ohaus ARA520 Adventurer Balance (Supplementary Fig. 6b). Gravimetric soil water content (θg) of the whole soil in the pots was calculated as

$$\theta g = \frac{Mwet - Mdry}{Mdry} \quad (2)$$

**Leaf feeding assay.** The stomatal conductance, transpiration and photosynthetic rate of the detached leaf feeding assay was recorded using either a LCpro-SD Portable Photosynthesis System (ADC Bioscientific) with 350 μmol m$^{-2}$ s$^{-1}$ light, 200 μmols s$^{-1}$ flow rate and 400 ppm CO$_2$ at 22 °C (Supplementary Fig. 1c, d) or

LI-COR LI-6400XT (LI-COR Biosciences) with 200 μmol m$^{-2}$ s$^{-1}$ light, 150 μmols s$^{-1}$ flow rate and 400 ppm CO$_2$ at 22 °C (Figs. 2c and 8b, d, f; Supplementary Figs. 2a, b and 17a, b, d, e, g, h). The detached leaf was fed with artificial xylem sap solution modified as described[68], containing 1 mM KH$_2$PO$_4$, 1 mM K$_2$HPO$_4$, 1 mM CaCl$_2$, 0.1 mM MgSO$_4$, 1 mM KNO$_3$, 0.1 mM MnSO$_4$, 1 mM K-H-malate, pH 6.0 (KOH) with or without GABA or muscimol supplement as indicated, detached leaves were pre-fed under 150 μmol m$^{-2}$ s$^{-1}$ light to allow the uptake of treatments for 45–60 min before recording. iWUE and WUE were calculated based on the equation as described in ref. [27].

**GABA measurement.** GABA concentration was determined using ultra performance LC (UPLC) as described previously[36]. Briefly, GABA was extracted from samples using 10 mM sodium acetate and derivatized with the AccQ Tag Ultra Derivatization Kit (Waters). Chromatographic analysis of GABA was performed on an Acquity UPLC System (Waters) with a Cortecs or Phenomenex UPLC C18 column (1.6 μm, 2.1 × 100 mm). The gradient protocol for amino acids analysis was used to measure GABA with mobile solvents AccQ Tag Ultra Eluents A and B (Waters). Standard GABA solution was used for calibration ranging from 0 to 150 μM. The results were analyzed by Empower chromatography software version 3 (Waters).

**GUS histochemical staining assays.** A GUS histochemical assay was performed using the methods described previously[69]. Three-to-four-week-old transgenic *pGAD2::GUS* plants were stained in buffer containing 50 mM Na phosphate pH = 7.0, 10 mM EDTA, 2 mM potassium ferrocyanide, 2 mM potassium ferricyanide, 0.1% (v/v) Triton X-100 and 0.1% (w/v) X-Gluc (5-bromo-4-chloro-3-indolyl β-D-glucuronide) during a 1.5 h incubation at 37 °C in the dark. The stained plants were destained in 70% ethanol. GUS-stained plants were imaged using an Axiophot Pol Photomicroscope (Carl Zeiss).

**Fluorescence microscopy.** The fluorescence of fluorescent proteins in transgenic *gad2-1/GC1::GAD2Δ-GFP*, *gad2-1/GC1::GAD2-GFP* and WT/*GC1::GAD2Δ-GFP* plants was imaged by confocal laser scanning microscopy using a Zeiss Axioskop 2 mot plus LSM5 PASCAL and argon laser (Carl Zeiss). Sequential scanning and laser excitation was used to capture fluorescence via the LSM5 PASCAL from GFP (excitation = 488 nm, emission band-pass = 505–530 nm), chlorophyll auto-fluorescence (excitation = 543 nm, emission long-pass = 560 nm). The fluorescence of fluorescent proteins in the mesophyll protoplasts of transgenic *almt9-2* complementation lines and *N. benthamiana* (Supplementary Fig 16c, d) was imaged using Nikon A1R Laser Scanning Confocal with DS-Ri1 CCD camera. Sequential scanning and laser excitation was used to capture fluorescence via the Nikon A1R Laser Scanning Confocal from GFP (excitation = 488 nm, emission = 525–575 nm), chlorophyll autofluorescence (excitation = 561 nm, emission = 595–645 nm).

**Reverse transcriptional PCR.** Reverse transcriptional PCR was determined by PCR amplification on cDNA synthesized from RNA extracted from plants as indicated. PCR amplified *GAD2*, *Actin2*, *GFP*, *GAD2 mRNA*, *UAS::GAD2Δ* and *ALMT9* using Phire Hot Start II DNA Polymerase (Invitrogen) with primer sets:
GAD2_rt_F (5′-ACGTGATGGATCCAGACAAAG-3′) and
GAD2_rt-R (5′-TACATTTTCCGCGATCCCT-3′);
Actin2_rt_F (5′-CAAAGGCCAACAGAGAGAAGA-3′) and
Actin2_rt_R (5′-CTGTACTTCCTTTCAGGTGGTG-3′);
GFP_rt_F (5′-GGAGTTGTCCCAATTCTTGTT-3′) and
GFP_rt_R (5′-CGCCAATTGGAGTATTTTGT-3′);
GAD2mRNA_rt_F (5′-ACGTGATGGATCCAGACAAAG-3′) and
GAD2mRNA_rt_R (5′-TCTTCATTTCCACACAAAGGC-3′);
UAS_GAD2_rt_F (5′-TCACTCTCAATTTCTCCAAGG-3′) and
UAS_GAD2_rt_R (5′-CGGCAACAGGATTCAATCTTAAG-3′);
ALMT9_rt_F (5′-AATACTCGAGAAACGGGGAGAG-3′) and
ALMT9_rt_R (5′-CATCCCAAAACACCTACGAAT-3′).

**Quantitative real-time PCR analysis.** Quantitative reverse transcription PCR was performed using KAPA SYBR FAST ABI PRISM kit (Kapa Biosystems) using a QuantStudio$^{TM}$ 12K Flex Real-Time PCR System (Thermo Fisher Scientific) to determine the expression levels of *GAD1*, *GAD2*, *GAD3*, *GAD4*, *GAD5*, *GABA-T*, *ALMT9*, *ALMT12*, *RD29A* and *RD22* genes with primer sets:
GAD1_qF (5′-TCTCAAAGGACGAGGGAGTG-3′) and
GAD1_qR (5′-AACCACACGAAGAACAGTGATG-3′);
GAD2_qF (5′-GTCTCAAAGGACCAAGGAGTG-3′) and
GAD2_qR (5′-CATCGGCAGGCATAGTGTAA-3′);
GAD3_qF (5′-CCGTTAGTGGCGTTTTCTCT-3′) and
GAD3_qR (5′-TCTCTTTGCGTCTCCTCTGG-3′);
GAD4_qF (5′-GTGTTCCGTTAGTGGCGTT-3′) and
GAD4_qR (5′GTCTCCTCTGGCGTCTTCTT-3′);
GAD5_qF (5′-TCAACCCACTTTCACTCTCA-3′) and
GAD5_qR (5′-TTCCTTCTCTTAGCCTCCTT-3′);
GABA-T_qF (5′-AGGCAGCACCTGAGAAGAAA-3′) and
GABA-T_qR (5′-GGAGTGATAAAACGGCAAGG-3′);

ALMT9_qF (5′-CAGAGAGTGGGCGTAGAAGG-3′) and
ALMT9_qR (5′-GGATTTGAAGGCGTAGATTGG-3′);
ALMT12_qF (5′-TTGACGGAACTCGCAGATAG-3′) and
ALMT12_qR (5′-CGATGGAGGTTAGAGCCAAG-3′);
RD29A_qF (5′-AAACGACGACAAAGGAAGTG-3′) and
RD29A_qR (5′-ACCAAACCAGCCAAGATGATT-3′);
RD22_qF (5′-AGGGCTGTTTCCACTGAGG-3′) and
RD22_qR (5′- CACCACAGATTTATCGTCAGACA-3′).
Expression levels of each gene was normalised to three control genes—*Actin2*,
*EF1α* and *GAPDH-A*—that were amplified with primer sets:
Actin2_qF (5′-TGAGCAAAGAAATCACAGCACT-3′) and
Actin2_qR (5′-CCTGGACCTGCCTCATCATAC-3′);
EF1α_qF (5′-GACAGGCGTTCTGGTAAGGAG-3′) and
EF1α_qR (5′-GCGGAAAGAGTTTTGATGTTCA-3′);
GAPDH-A_qF (5′-TGGTTGATCTCGTTGTGCAGGTCTC-3′) and
GAPDH-A_qR (5′-GTCAGCCAAGTCAACAACTCTCTG-3′).

**Reporting summary**. Further information on research design is available in the Nature
Research Reporting Summary linked to this article.

## Data availability

Sequence data used in this paper can be found in The Arabidopsis Information Resource
database (https://www.arabidopsis.org/) under the following accessions: *GAD1*
(At5g17330), *GAD2* (At1g65960), *GAD3* (At2g02000), *GAD4* (At2g02010), *GAD5*
(At3g17760), *GABA-T* (At3g22200), *ALMT9* (At3g18440), *ALMT12* (At4g17970),
*RD29A* (At5g52310) and *RD22* (At5g25610). Other data that support the findings of this
study are available from the corresponding author upon request. Source Data are
provided with this paper.

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

## Acknowledgements

The authors would like to thank Dr Katja Bärenfaller from the University of Zurich for providing JR11-2 (Col-0) seeds; Prof. Zhonghua Chen from Western Sydney University for assisting with barley epidermal assays; Dr Cornelia Eisenach from the University of Zurich for providing *ALMT9* constructs, *almt12* and *almt9* seeds and Prof. Stephen D Tyerman, Prof. Enrico Martinoia and Dr Alexis De Angeli for valuable discussions. This work was funded by Waite Research Institute, funding to M.G. and B.X., ARC Discovery grants DP170104384 and DP210102828 to M.G. and R.H. and ARC Centre of Excellence (CE140100008) and Grains Research and Development Corporation funding (UWA00173) to M.G. Author X.F. was supported by supported by a Chinese Scholarship Council. X.Z. was supported by a Chinese Scholarship Council Travelling Fellowship.

## Author contributions

B.X. constructed all materials and performed all experiments except the following: Y.L. generated *almt9/almt12* mutants and performed experiments on stomatal aperture assays treated with hydrogen peroxide and calcium and of *almt12* and *almt9/12* mutants. X.F. generated *almt9* complementation lines, performance aperture and conductance measurement of complementation plants. X.Z. performed all non-*Arabidopsis* aperture measurements, except for barley performed by N.S. Author L.C. performed GABA measurements supervised by M.O. Author A.B. performed ABA quantification supervised by E.J.E. Author J.H. acquired images used in Fig. 1. M.G., B.X. and R.H. conceived the research. B.X. drafted all figures. M.G. and B.X. drafted the manuscript. All authors provided edits.

## Competing interests

The authors declare no competing interests.
