## [Peer Review File · Nature Communications]

Reviewers' comments:

Reviewer #1 (Remarks to the Author):

GABA is a non-protein amino acid which accumulates in plants upon stress. With evidence suggesting that GABA regulates the activity of Aluminum-activated Malate Transporters (ALMTs), Xu et al investigated a possible role for GABA-regulation of stomatal movements. Exogenous treatments in leaf epidermal peels with GABA (and analog) suggest that GABA has a negative effect on both stomatal opening and stomatal closure. GAD2 is the major GABA synthesis gene in leaves. Mutants *gad2*, which accumulate less GABA, have more opened stomata and are more susceptible to drought. The authors also investigated whether GABA could affect the activity of two ALMTs, 9 and 12, and propose a model in which GABA inhibits stomatal opening and stomatal closing by inhibiting guard cell expressed ALMTs.

Although the hypothesis and mechanism explored in this work are new and potentially interesting, the model and phenotypes seem not consistent with the data in relevant results and the quality of some data is in need of improvement.

Major comments

1- A main concern about this work lies on the proposed GABA mechanism of action. Exogenous GABA treatment in leaf epidermal peels suggest that GABA inhibits both stomatal opening and closing. These findings are supported by analyses of *almt* mutants (9 and 12). On the other hand, *gad2* mutants have more opened stomata and higher steady-state stomatal conductance. Also, dark-induced stomatal closure in *gad2* mutants is enhanced as shown in figure 4e and stomatal apertures levels are also comparable between *gad2* mutant and WT under dark treatment (e.g. Suppl. 4i, control boxes). If mutants with less GABA are supposedly more responsive to stimulus-induced stomatal closure, these mutants should have either WT-like response to drought if not even be more drought tolerant. In figure 5, plants transformed with constitutively active GAD2 are more drought tolerant. If these plants accumulate more GABA, they should have impaired stomatal closure responses. I have not found a reasonable explanation as to why *gad2* mutants failed to close their stomata in response to drought.

2- Does the exogenously added GABA actually enter guard cells as thought in the experiments? Might something else explain the effects on stomatal opening and closing?

3- Drought experiments in figure suppl 8 do not contain data on soil moisture levels or pot weight. The authors must ascertain that the water-limiting stress is replicated in all genotypes and all individual plants being compared. One has to make sure that all genotypes are exposed to the same reproducible stress severity. Without measuring and equalising soil moisture or weighting pots and adjusting weights to be the same by adding water, drought experiments as reported are considered to be of little meaning and robustness, as plant scientists have come to realise. Using methods without weighing pots have been critiqued in the literature for several years now and protocols for stringent tests have been published by several labs.

4- In figure 1, the stomata become fully opened after 80 minutes of exposure to light. On the other hand, in Figure 4d, steady-state stomatal conductance is achieved within ~20 minutes of exposure to light. This time discrepancy suggests that the leaf epidermal peels might not be able to fully respond to external stimuli and that more accurate measurements need to be obtained by using time-resolved stomatal conductance measurements.

5- Time-resolved stomatal conductance data would have been needed for experiments implying the roles of ALMT9 and GAD2 (Figure 6 and 7) and for ALMT12 (Figure 8).

6- In Figure 7e, the *almt9-2* mutant line would be required for comparison.

7- The data showing that GABA increases in water-stressed leaves (Suppl Fig 4C) seems variable and more data is needed. The changes in water-stressed GABA concentrations in Fig. 3A WT and Suppl Fig 4C are distinct even though the conditions seem to be the same. More experiments are needed to ascertain this conclusion.

8- The title and abstract are speculative and overstate the results. Stress "memory" was not determined and "optimisation" was not analysed.

9- What were the criteria to define the concentration of GABA and analog to be used on distinct plant species? Is there a dose-dependency trend in exogenous treatments?

10- Line 172 "...plants have similar leaf RWC...(Figure S 7e-h). There is no data on Relative Water Content in this Figure?

10- In figure 4, GC1:GAD2Δ transformed in gad2 mutants rescue the stomatal phenotype to WT levels but, in the WT background, the same construct is interpreted to make plants more drought tolerant. Are the GABA levels different between these two genotypes?

11- Figure Suppl. 1 C, D and E show Muscimol-inhibition of stomatal opening. Similar data should be included to test GABA and Muscimol-inhibition of stomatal closure.

12- Figure Suppl. 2 – Authors should test whether stomatal opening is not affected too.

Other comments:

The authors show GABA-rescue of gad2 mutant phenotype in stomatal aperture measurements. Gas exchange data are needed (e.g. Figure 4 d and e) to test whether GABA can rescue the phenotype in time-resolved experiments.

In Suppl Figure 2: Include labels (clear vs. dark blocks).

In Suppl Figure 8: Include labels (black, blue and red boxes).

Line 157 – Figure 4a shows expression and GABA.

Line 206 – Figs Suppl 9a-d do not show GABA levels.

Line 245- Why is figure 1 been referred to?

Line 247 – Figure 8a shows dark to light and not the opposite.

Line 249 – 8b refers to light to dark treatment.

Add page numbers including in supplement.

Reviewer #2 (Remarks to the Author):

In this manuscript, Xu et al. investigated the role of gamma-aminobutyric acid (GABA) in plant for controlling light- and dark-induced stomatal movement. The non-protein amino acid, GABA is known to accumulate in plants in response to various environmental stresses including drought. The same laboratory previously reported that plant aluminium-activated malate transporters (ALMTs) have a putative GABA-binding motif similar to that of well-known mammalian GABA receptors, and GABA directly regulates the ALMT activity (Ramesh et al., 2015 Nat Commun). In this study, Xu et al. show that GABA antagonizes light- and dark-induced stomatal movement. The stomatal response to GABA is conserved in various plant species. They also provide evidence that two guard cell-expressed ALMTs ALMT9 and ALMT12 are involved in the guard cell GABA signaling. The authors' finding is interesting and would help to understand the role of GABA in plants. The reviewer thinks that additional experiments are required to fully support authors' hypothesis (especially comments 1 and 2).

(1) The major concern of this manuscript is that GABA regulation of AtALMT9 and AtALMT12 activity has been not proven by electrophysiological analysis, although the same group previously showed that several other ALMT transporters are GABA sensitive (Ramesh et al., 2015). This can be performed easily using two-electrode voltage-clamp (TEVC) electrophysiology as their previous work (Ramesh et al., 2015).

(2) GABA sensitivity of stomatal opening in *almt9* mutant was not restored by expression of ALMT9F243C/Y245C that has mutations in the putative GABA binding region (Figure 7). Similar mutant analysis should be performed for ALMT12. In addition, the GABA sensitivity of ALMT9F243C/Y245C and the ALMT12 mutant also needs to be analyzed using TEVC electrophysiology.

(3) AtALMT12 is involved in not only dark-induced stomatal closure but also abscisic acid (ABA)- and CO₂-induced stomatal closure (Meyer et al., 2010 Plant J; Sasaki et al., 2010 Plant Cell Physiol). Therefore, the authors need to check whether GABA can antagonize stomatal closure induced by other stimuli such as ABA. This experiment would help well to understand mechanism and physiological function of guard cell GABA signaling at a deeper level.

(4) Considering the localization and membrane topology of ALMT9 and ALMT12, the two ALMTs might sense GABA at different cellular location. (Vacuole GABA targets ALMT9 and apoplasmic GABA targets ALMT12?) Please mention and discuss this point.

Minor comment

(1) Please add scale bars to stomatal pictures of Supplementary Figure 3.

(2) L262: "ALMT" -> "ALMTs".

Reviewer #3 (Remarks to the Author):

GABA is a major metabolite in plants, which has been studied for decades. It is synthesized in the cytosol, and functions in primary carbon/nitrogen metabolism and in respiration. Accordingly, the concentration of GABA in plant cells and in specific subcellular compartments is among the highest concentration of all amino acids, in the mM range. In addition, plant GABA was suggested to have functions as an osmoregulator (similar to proline) and pH regulation. GABA synthesis is enhanced by various stressors, mostly by activation of cytosolic glutamate decarboxylase (GAD), which is highly responsive to acidic pH, and to Ca²⁺/calmodulin at neutral pH. In Arabidopsis there are 5 genes encoding GAD enzymes, some with tissue specific expression. For example, GAD1 is mostly expressed in roots. A major question over the years has been the possible role of GABA as a signaling molecule, similar to its in animals. In recent years, the group of the corresponding author has shown that GABA can function as a negative regulator of anion flux through ALMT malate transporters. Yet, its role as a signaling molecule in stress responses has not been demonstrated.

In the submitted manuscript the authors suggest a role for GABA in controlling stomata aperture, transpiration and drought tolerance, via inhibition of plasma membrane and tonoplast-localized ALMT anion transporters. My review is divided in two parts. In the first part (A) I assess the results concerning the effects of GABA synthesis on stomata aperture, transpiration rate and drought tolerance. In the second part (B) I review the evidence concerning GABA function as a signaling molecule (rather than as a metabolite) in guard cells.

A. The results concerning GABA effects on stomata aperture in excised epidermal strips are consistent with the results obtained in intact plants regarding GABA-dependent transpiration rates and drought tolerance, demonstrating the importance of GABA. This is nicely shown by comparing wild type with *gad2* mutants and to complementation lines expressing GAD2 Δ C (presumably a constitutively active form of the enzyme) under the transcriptional regulation of a guard cell-

specific promoter. I find this molecular – physiological part convincing, novel and worthy of publication. I do have a minor reservation concerning the authors conclusion that only GAD2 is operating in these processes. In lines 136-138 the authors describe the results presented in Supplementary Fig. 5 suggesting that transcript levels of GABA-related genes “were similar in wild type and *gad2* lines”. I find this statement somewhat inaccurate, particularly on day 7 of the drought treatment. Similarly, the authors state that ABA-marker genes are expressed similarly in WT and *gad2* mutants. This is also inaccurate, particularly on day 7 of the drought treatment of mutant *gad2-1*.

B. Is GABA regulating stomata aperture, transpiration and drought tolerance by its role as a signaling molecule or as a major metabolite (e.g. in respiration, osmoregulation, etc.), or both? As the first step to assess if GABA functions as a signaling molecule, the authors identified ALMT anion transporters as possible targets of GABA in guard cells. The authors rely on their previous work showing that ALMT transporters contain a GABA-binding domain which negatively regulates anion transport when GABA binds. The authors here focused on ALMT9, a vacuolar anion channel essential for proper regulation of stomata opening, and on the plasma membrane ALMT12, essential for proper stomata closing. Using mutants of both transporters, the authors nicely show the effects of GABA on the opening of stomata via ALMT9 and of closing stomata via ALMT12. Subsequently, in order to assess whether GABA affects stomatal conductance by binding to the expected site on ALMT9, the authors performed complementation of an *almt9* mutant with WT ALMT9 (Fig. 7e), or with a specific double mutation of ALMT9 (F243C/Y245C), which is expected to prevent the regulation by GABA, according to a previous publication of the corresponding author. However, the results are not perfect. The authors tested only two complementation lines (#1 and #2) for each construct. Of the two lines with the WT-expressed protein, complementation line ALMT9#1 did not restore the response to GABA (it should have) and complementation line ALMT9#2 restored an effect of GABA at a rather low statistical confidence ($p < 0.05$). Therefore, the fact that the two complementation lines with the mutations did not restore a response to GABA is not convincing enough for concluding an effect of the mutation on responsiveness to GABA. As this set of experiments is critical for drawing the main conclusion of the submitted manuscript, I believe that the authors should test more complementation lines (of WT and mutant) to provide more convincing data.

Another issue is related to the fact that recently ALMT1 from wheat was shown (by the corresponding author and colleagues) to function as a GABA transporter (Ramesh et al., 2018) and that mutation F213C (analogous to F243C in *Arabidopsis* ALM9) prevented GABA influx and efflux through the transporter (Ramesh et al., 2018). Hence, if the double mutation (F243C/Y245C) impedes GABA flux through the tonoplast and plasma membrane ALMTs, and consequently affects GABA subcellular compartmentalization in guard cells, it is possible that this effect is the actual driver of stomata aperture modulation under water deficit conditions. The authors should address this issue. Furthermore, the lack of an *in vivo* reporter for GABA, to determine real-time spatial and temporal GABA concentrations in different cellular compartment in the context of the stress response, makes it difficult to understand what really happens with GABA in guard cells. The authors determined whole leaf GABA levels in plant extracts but these cannot be interpreted as the levels of GABA in specific leaf cell types, or in specific cellular compartments.

Minor point

It is not clear to me why the authors chose to refer to the GABA signal as a “memory” signal. In order to determine that a substance acts like a memory signal, it would be necessary to determine its effects on subsequent stress responses, after the first stress has been alleviated for some time. In the submitted study the authors have not attempted this type of experiments, but rather studied the effects of GABA during a given stress response and immediately upon recovery from the stress. Therefore, I don’t find a justification to refer to this case as “memory”.

Recommendation: The authors describe a novel mechanism of GABA controlling stomata aperture, transpiration and drought tolerance. These findings should be published in Nature Communication, pending revision.

Response to reviewers.

We would like to thank the reviewers for the favourable remarks about our work, noting all reviewers commented positively on the novelty of GABA as a plant signal and indicated its suitability in a revised form for *Nature Communications*. We have now revised the manuscript in relation to where clarifications were sought and have been fortunate enough to have some limited ability, over the last 6 months, to conduct additional experiments that strengthen our findings and the weight of our conclusions. As you can see this is a substantial piece of work with 138 sub figures in total that underpin our conclusions, and this has been 8 (full-time) years in the making. We are confident that the body of work now presented is of sufficient maturity, robustness and substance to warrant acceptance. We are also proud that our work raises multiple new questions that lend themselves to substantial future research efforts to fully decipher GABA signalling in guard cells and elsewhere in the plant in years to come, and we kindly request that we are not asked to delve deeper into this realm with further experiments, but rather this is left for the confines of future discrete work. We hope that we have been able to adequately address all reviewer concerns and it is now deemed appropriate for the current manuscript to be published in *Nature Communications*.

All reviewer comments are found in *italics* below. We delimit our responses with ****, ****.

Reviewers' comments:

Reviewer #1 (Remarks to the Author):

GABA is a non-protein amino acid which accumulates in plants upon stress. With evidence suggesting that GABA regulates the activity of Aluminum-activated Malate Transporters (ALMTs), Xu et al investigated a possible role for GABA-regulation of stomatal movements.

Exogenous treatments in leaf epidermal peels with GABA (and analog) suggest that GABA has a negative effect on both stomatal opening and stomatal closure. GAD2 is the major GABA synthesis gene in leaves. Mutants gad2, which accumulate less GABA, have more opened stomata and are more susceptible to drought. The authors also investigated whether GABA could affect the activity of two ALMTs, 9 and 12, and propose a model in which GABA inhibits stomatal opening and stomatal closing by inhibiting guard cell expressed ALMTs.

Although the hypothesis and mechanism explored in this work are new and potentially interesting, the model and phenotypes seem not consistent with the data in relevant results and the quality of some data is in need of improvement.

Major comments

1- A main concern about this work lies on the proposed GABA mechanism of action. Exogenous GABA treatment in leaf epidermal peels suggest that GABA inhibits both stomatal opening and closing. These findings are supported by analyses of almt mutants (9

and 12). On the other hand, *gad2* mutants have more opened stomata and higher steady-state stomatal conductance. Also, dark-induced stomatal closure in *gad2* mutants is enhanced as shown in figure 4e and stomatal apertures levels are also comparable between *gad2* mutant and WT under dark treatment (e.g. Suppl. 4i, control boxes). If mutants with less GABA are supposedly more responsive to stimulus-induced stomatal closure, these mutants should have either WT-like response to drought if not even be more drought tolerant.

****The reviewer makes a valid point, and we thank them for requiring us to re-evaluate and re-frame our findings.

As a result of this re-evaluation, we have made structural changes to the manuscript to concentrate on the finding that GABA improves water use efficiency and subsequently drought tolerance through reduced stomatal pore opening, and that this is transduced via inhibition of ALMT9 activity (the tonoplast-localised opening-associated anion channel). Our data clearly show *gad2* leaves (with low GABA concentrations) have stomata that are open to a greater extent and lose more water than WT leaves. This leads to reduced water use efficiency and decreased drought tolerance (see Figure 3, 4). The use of knockout plants is instructive, but so are our results generated from *GAD2* overexpression, as we show that increasing GABA concentration in guard cells of wildtype plants reduces the opening of guard cells and water loss (Figure 5) – this is highly significant as it is a new route for improving plant water use efficiency.

To further illustrate the dominant effect that GABA has on modulating stomatal opening we conducted several new experiments. In new Figure 2c-d & 8b-g and Supplemental Figure 2&15, the clear effect that GABA has on ALMT9 is shown by observing real time stomatal conductance during opening on wildtype, *almt9* and complemented mutant plants (also see reviewer 1, response 5). If the GABA effect on ALMT9 did not override any inhibitory effect that GABA may have on stomatal closure, then the reviewer would indeed be correct in their assertion that *gad2* plants should have WT-like, or better, drought tolerance. More than this though, it appears that GABA in these excised leaves fed GABA inhibits only stomatal opening but not stomatal closure. This fact is now clearly stated in the MS.

As such, GABA does not appear to inhibit closure in intact leaves under the conditions tested, and does not appear to be a physiological signal during drought. Therefore, we have decreased the emphasis on the effect of GABA on closure in epidermal peels in the new manuscript.

It is also instructive to specifically address these two points made by reviewer 1:****

- *dark-induced stomatal closure in gad2 mutants is enhanced as shown in figure 4e*

****We agree the rate of closure is enhanced in *gad2*, but as can be seen from Figure 4d, *gad2* stomatal conductance is higher in the light and is reduced (over the short time period) to a value that is still greater than wildtype or the complemented mutant. Water loss of *gad2* is therefore still greater in both the light and the dark.****

- *and stomatal apertures levels are also comparable between gad2 mutant and WT under dark treatment (e.g. Suppl. 4i, control boxes).*

****This is incorrect. In Suppl. Figure 4i (now Suppl. Figure 6i), after a 2h dark treatment the stomatal apertures of *gad2-1* and *gad2-2* are greater than in wildtype plants. We have now indicated the significance on the graphs. Over this time period this would mean more water is lost from *gad2* mutants than wildtype plants, even in the dark.

In summary, we concede that our focus in the original manuscript on both ALMT9 and ALMT12 may lead to some confusion. Due to our new data and re-examination of previous findings that show that GABA does not inhibit closure in intact leaves - in the conditions used in our experiment - we have decided to remove the data on ALMT12 in the manuscript. Whilst our previously shown preliminary data suggests that GABA action on ALMT12 influences stomatal closure in epidermal peels, we have not yet been able to ascribe a physiological role for this observation; certainly, ALMT12 does not appear to play a role with respect to drought and WUE in this study. We therefore think it is premature to include this data as it is merely an interesting distraction to the main point of the manuscript. We instead suggest in the discussion that examining the ALMT12-GABA interaction should be the subject of further work and also remove ALMT12 from our final model (new Figure 10). We instead more clearly emphasise the role of GABA in reduced stomatal opening and improving water use efficiency.****

In figure 5, plants transformed with constitutively active GAD2 are more drought tolerant. If these plants accumulate more GABA, they should have impaired stomatal closure responses. I have not found a reasonable explanation as to why gad2 mutants failed to close their stomata in response to drought.

**** Overexpression of *GAD2* in wildtype plants leads to greater drought tolerance. The reason for this is that stomata open less each day, and the plants lose less water. As can be seen from feeding GABA to leaves, GABA does not prevent closure from occurring (new Figure 2c). Therefore, GABA reduces opening but not the degree of stomatal closure. This can also be seen in the *gad2* complemented plants (Figure 4c,d); *GAD2* expression in the guard cell still results in a reduction in stomatal conductance to the same degree as wildtype (Figure 4d). The *gad2* phenotype is opposite to the *GAD2* overexpression phenotype in wildtype plants. Our evidence suggests *gad2* mutants lose their water more quickly than wildtype plants because ALMT9 is deregulated. Stomatal aperture is a consequence of both stimulation of closure and inhibition of opening. With ALMT9 not fully inhibited in *gad2*, as there is a lack of GABA, apertures do not decrease as much as in wildtype plants during closure (Figure 4d). We have never claimed that *gad2* mutants do not respond to drought signals. For the referees' convenience, we included a new data set that shows that *gad2* mutants are able to respond to both low and high ABA (new Supplementary Figure 13g-j), though their aperture is larger than that of WT at all times (due to lack of ALMT9 inhibition).

We trust that our restructuring of the manuscript, and new data, is sufficient to address this point.****

2- Does the exogenously added GABA actually enter guard cells as thought in the

experiments? Might something else explain the effects on stomatal opening and closing?

****Our data is consistent with GABA entering the cytosol of guard cells. The pertinent point here is that we clearly demonstrate that the presence of cytosolic GABA leads to inhibition of opening. Several lines of evidence suggest that this is the case:

- Our new leaf feeding data shows GABA inhibits opening via ALMT9 (Figures 2c and 8b).
- ALMT9 is on the tonoplast. For GABA to inhibit ALMT9 it would have to act through the cytosol.
- We are aware of a number of GABA permeable transporters in plants resident on the plasma membrane including ProT and even ALMTs.
- Our recent work suggests that the GABA site of action on ALMTs is from the cytosol (Long, Tyerman & Gilliam, 2020, New Phytologist 225:671-678). One of the diagnostics for this work was the ability of the membrane impermeable GABA-BODIPY conjugate to block ALMT activity from only the cytosolic side of the membrane, not the extracellular side. Based on these findings we have now conducted a new experiment using the membrane impermeable muscimol-BODIPY conjugate on epidermal peels (new Supplementary Figure 12). This experiment shows that, unlike muscimol, muscimol-BODIPY does not inhibit stomatal opening or closure when applied from the extracellular solution.
- Guard cell specific, cytosolic expression of *GAD2delta* results in complementation of *gad2* and reduction in water use of wildtype background plants.
- Guard cell specific cytosolic expression of full-length *GAD2* cannot rescue the higher stomatal conductance of the *gad2* mutant, but 35S::*GAD2* does, suggesting GABA generated from other cell types contributes to the signalling and enter guard cells to coordinate gas exchange.
- Spongy mesophyll cell specific expression of *GAD2delta* does not result in complementation of *gad2* plants as the guard cell cytosol, most likely due to insufficient supply from the mesophyll alone.

We don't possess any other data that suggests GABA is inhibiting stomatal opening through any other mechanism than through transduction of a cytosolic GABA signal through ALMTs. We have therefore expanded the results section relating to cell specific effects to more comprehensively cover this topic. ****

3- Drought experiments in figure suppl 8 do not contain data on soil moisture levels or pot weight. The authors must ascertain that the water-limiting stress is replicated in all genotypes and all individual plants being compared. One has to make sure that all genotypes are exposed to the same reproducible stress severity. Without measuring and equalising soil moisture or weighting pots and adjusting weights to be the same by adding water, drought experiments as reported are considered to be of little meaning and robustness, as plant scientists have come to realise. Using methods without weighing pots have been critiqued in the literature for several years now and protocols for stringent tests have been published by several labs.

****We apologise for omitting this detail in the methods. We followed the standard protocols as outlined above (equalising soil weights, and watering to weight). This detail is now included in the methods.****

4- In figure 1, the stomata become fully opened after 80 minutes of exposure to light. On the other hand, in Figure 4d, steady-state stomatal conductance is achieved within ~20 minutes of exposure to light. This time discrepancy suggests that the leaf epidermal peels might not be able to fully respond to external stimuli and that more accurate measurements need to be obtained by using time-resolved stomatal conductance measurements.

****The reviewer is correct. It is perhaps not as widely acknowledged as it should be in the literature that stomatal aperture data from epidermal peels does not always fully replicate what can be observed in intact or excised leaves.

However, we have shown in this manuscript that the results obtained with respect to the inhibition of opening by GABA (epidermal peels, excised leaves and intact plants via genetic complementation are consistent). Therefore, with respect to GABA inhibition of opening, we are confident in being able to state our conclusions – and the validity of the epidermal peel experiments. We concede the rate of opening in excised strips and intact plants is not identical and this has been shown by others previously. However, we have now generated time resolved datasets with GABA fed through the petiole (new Fig 2c; Supplementary Figure 2a-b) that is a clear demonstration of the inhibitory effect that GABA has on stomatal opening and its corollary – the inhibition of stomatal conductance and the increase in water use efficiency. This new data is another clear demonstration that GABA inhibits the rate of opening.

We again thank the reviewer for these comments as it made us review our data and conduct new experiments that show closure is not inhibited by GABA in intact leaves (see response to comments 1, reviewer 1, for further discussion). We have therefore reduced our emphasis on GABA inhibition of closure in the manuscript and removed the data relevant to ALMT12.****

5- Time-resolved stomatal conductance data would have been needed for experiments implying the roles of ALMT9 and GAD2 (Figure 6 and 7) and for ALMT12 (Figure 8).

****Although we were willing to argue that the steady state conductance already shown for all the mutants coupled to the time resolved conductance's of wildtype, *gad2* and *gad2* complemented plants undergoing a transition was sufficiently illustrative of our model in the original manuscript, we were keen to definitively demonstrate that GABA action on ALMT9 was the major mechanism at play. We fully accept that the data outlined in original Figure 7e and f (now Figure 8a, Figure 9) could have been stronger. Therefore, we repeated several assays and included a new dataset.

We now show time resolved stomatal conductance of wildtype (Figure 2), *almt9*, *almt9* complemented with *ALMT9* and *almt9* complemented with a functional *ALMT9* anion transporter with mutations in the putative GABA responsive site (Figure 8). These data clearly shows that GABA inhibits opening of WT and *almt9/ALMT9* stomata, but not in *almt9*

or the *almt9/ALMT9doublepointmutant*. This is very strong evidence to suggest that GABA acts on ALMT9 to reduce stomatal opening. We have removed data on ALMT12 as explained above.****

6- In Figure 7e, the *almt9-2* mutant line would be required for comparison.

****We apologise for omission, we now include *almt9-2* in Figure 8a (with more stomata counted). We repeated this experiment two more times, and conducted the additional leaf feeding assay on these lines outlined above. All data is consistent with GABA inhibiting ALMT9 activity.****

7- The data showing that GABA increases in water-stressed leaves (Suppl Fig 4C) seems variable and more data is needed. The changes in water-stressed GABA concentrations in Fig. 3A WT and Suppl Fig 4C are distinct even though the conditions seem to be the same. More experiments are needed to ascertain this conclusion.

****We have repeated this experiment on wildtype plants 6 times in this manuscript and, on each occasion, there is a detectable increase in GABA concentration upon drought. The exact values do change as would be expected in independent biological experiments. We also get biological variation amongst replicates. But in every experiment, we detect a statistically significant increase in mean GABA concentration upon drought. In Supplementary Figure 6c we plot a developmental control which shows that GABA does not just build up in control conditions over time. We are not the only group to detect drought or stress induced increases in GABA – see e.g. review by Ramesh et al., 2016, CMLS 74:1577-1603.****

8- The title and abstract are speculative and overstate the results. Stress “memory” was not determined and “optimisation” was not analysed.

****We agree the title was speculative. We have rephrased the title to: GABA signalling modulates stomatal opening to enhance plant water use efficiency and drought resilience.****

9- What were the criteria to define the concentration of GABA and analog to be used on distinct plant species? Is there a dose-dependency trend in exogenous treatments?

****We applied concentrations found in the physiological range, and at a level measured in the tissue under drought, they are also those determined as active on the plant system through previous work Ramesh et al. 2015 Nature Communications. In the future we agree, it would be good to determine the dose dependency but we request that this is left for future studies based on the strength and depth of our findings.****

10- Line 172 “...plants have similar leaf RWC...(Figure S 7e-h). There is no data on Relative Water Content in this Figure?

****We apologise for the omission; this was accidentally removed prior to submission and this has now been reinstated into new Supplementary Figure 11e. ****

10- In figure 4, *GC1:GAD2Δ* transformed in *gad2* mutants rescue the stomatal phenotype to WT levels but, in the WT background, the same construct is interpreted to make plants more drought tolerant. Are the GABA levels different between these two genotypes?

The whole tissue GABA was marginally higher in the WT background, this is now shown in Figure 5b. *

11- Figure Suppl. 1 C, D and E show Muscimol-inhibition of stomatal opening. Similar data should be included to test GABA and Muscimol-inhibition of stomatal closure.

This has been performed and is now included as new Figure 2c and Supplementary Figure 2a-b.

12- Figure Suppl. 2 – Authors should test whether stomatal opening is not affected too.

****We have now performed these experiments, stomatal opening can occur normally after washout of GABA (Supplemental Figure 4a-b).****

Other comments:

*The authors show GABA-rescue of *gad2* mutant phenotype in stomatal aperture measurements. Gas exchange data are needed (e.g. Figure 4 d and e) to test whether GABA can rescue the phenotype in time-resolved experiments.*

****We already show this using genetic complementation (Figure 4d,e). We therefore don't think this is necessary to show using GABA feeding.****

In Suppl Figure 2: Include labels (clear vs. dark blocks).

In Suppl Figure 8: Include labels (black, blue and red boxes).

Line 157 – Figure 4a shows expression and GABA.

Line 206 – Figs Suppl 9a-d do not show GABA levels.

Line 245- Why is figure 1 been referred to?

Line 247 – Figure 8a shows dark to light and not the opposite.

Line 249 – 8b refers to light to dark treatment.

Add page numbers including in supplement.

****Thanks for pointing out these issues, all changes have been made.****

Reviewer #2 (Remarks to the Author):

In this manuscript, Xu et al. investigated the role of gamma-aminobutyric acid (GABA) in plant for controlling light- and dark-induced stomatal movement. The non-protein amino acid, GABA is known to accumulate in plants in response to various environmental stresses including drought. The same laboratory previously reported that plant aluminium-activated malate transporters (ALMTs) have a putative GABA-binding motif similar to that of well-known mammalian GABA receptors, and GABA directly regulates the ALMT activity (Ramesh et al., 2015 Nat Commun). In this study, Xu et al. show that GABA antagonizes light- and

dark-induced stomatal movement. The stomatal response to GABA is conserved in various plant species. They also provide evidence that two guard cell-expressed ALMTs ALMT9 and ALMT12 are involved in the guard cell GABA signaling. The authors' finding is interesting and would help to understand the role of GABA in plants. The reviewer thinks that additional experiments are required to fully support authors' hypothesis (especially comments 1 and 2).

(1) The major concern of this manuscript is that GABA regulation of AtALMT9 and AtALMT12 activity has been not proven by electrophysiological analysis, although the same group previously showed that several other ALMT transporters are GABA sensitive (Ramesh et al., 2015). This can be performed easily using two-electrode voltage-clamp (TEVC) electrophysiology as their previous work (Ramesh et al., 2015).

****These experiments are actually not as straightforward as the reviewer suggests. We of course did attempt them - for at least a solid 12 months by two postdocs (in both the Gilliam and Hedrich labs), but for a variety of reasons we request that they are not a pre-requisite for publication as follows:

- 1) Functional characterisation in heterologous systems is commonly known to inadequately replicate what occurs *in planta* due to the absence of regulatory factors. This can be an advantage when reconstituting signalling pathways but is a critical drawback when not all of the factors are known. An illustrative example is the characterisation of the major plant guard cell closure anion channel (the S-type) in *Xenopus laevis* oocytes. When the first account of the molecular identity of S-type anion channel was published in 2008, in two papers back-to-back in *Nature*, no S-type anion channel attributable currents could be obtained in oocytes following injection with *SLAC1* cRNA. In the subsequent 12 years, and many high-profile papers later, the ABA-activated pathway has been slowly reconstituted in heterologous systems including several interacting proteins, including a recent paper in *Nature Communications* that shows a previously unidentified interacting kinase (MPK3) (Takahashi et al., 2020 *Nature Communications* **11**, 12). We would like to re-state that in our manuscript we are the first to demonstrate that GABA is an *in planta* signalling molecule. This is not just a single signalling element in a well characterised signalling pathway, but the first identification of a whole new signalling pathway.
- 2) We were unable to detect ALMT9 activity in *Xenopus laevis* oocyte plasma membranes despite earlier reports (Kovermann et al., 2007, *Plant Journal*).
- 3) Unlike de Angeli et al., 2013 *Nature Communications* 4:1804, we were unable to detect stable ALMT9 activity in isolated tobacco mesophyll cells following transient expression in both the whole vacuole mode and isolated membrane patches. We were able to successfully transform tobacco cells and detect malate induced chloride currents which we could ascribe to ALMT9, but we encountered rundown within seconds, making any experiments to determine the effects of GABA impossible to interpret. It should be remembered that the tobacco mesophyll vacuole is not the native home of Arabidopsis ALMT9, and with all the cytoplasm being absent such experiments are difficult to interpret - even positive results.
- 4) Functional characterisation via electrophysiology *in planta* (or isolating tonoplasts) is very difficult to interpret for ALMT9 due to the magnitude of the currents and other

background currents. In fact, we attempted to study the GABA sensitivity of anion currents across the tonoplast of wildtype and *almt9* mutants by patch clamping isolated guard cell vacuoles. But these were unsuccessful as we were unable to resolve currents that we could ascribe to ALMT9 due to the small size of the organelle and ALMT9 density. So far there is single Arabidopsis guard cell vacuole ion channel patch clamp study published by the co-author Hedrich lab (Rienmüller et al. 2010). In the latter one SV channel TPC1 could be well resolved, because TPC1 is most prominent vacuole ion channel with respect to both unitary conductance and density.

- 5) We have removed ALMT12 from this manuscript as it is clearly not the target for improving water use efficiency, so it was not appropriate to include such functional characterisation for this protein here.
- 6) We are confident our genetic complementation work more adequately portrays the regulation of these guard cell localised ALMT than can be obtain in heterologous systems at this stage. This includes the use of an ALMT9 point mutant that is an active but non-GABA responsive channel, which results in greater stomatal conductance than when complementing with ALMT9 (new Fig 9, with new data included). Only with substantial further work, out of the scope of this manuscript, will we be able to discover the co-factors that are needed for ALMT9 to function as it does *in planta* – for SLAC1, it has been 12 years after its discovery and new co-factors are still being identified.

We have also added the following statements in the results and discussion, acknowledging the difficulties in such experiments but the future need:

L292 – ‘We examined the potential regulation of ALMT9 by GABA by focusing solely on *in planta* studies as it is difficult to faithfully replicate regulatory pathways from guard cells in heterologous systems^{e.g. 41-46}

L394 - ‘Such studies will be able to resolve questions such as whether GABA can act directly on guard cell ALMTs, as appears to occur for wheat ALMT1^{18, 39}, or whether other signalling intermediates are also involved.’

(2) GABA sensitivity of stomatal opening in almt9 mutant was not restored by expression of ALMT9F243C/Y245C that has mutations in the putative GABA binding region (Figure 7). Similar mutant analysis should be performed for ALMT12. In addition, the GABA sensitivity of ALMT9F243C/Y245C and the ALMT12 mutant also needs to be analyzed using TEVC electrophysiology.

****As outlined in the response to reviewer 1. We have removed our focus on ALMT12 in this manuscript. ALMT9 is the protein responsible for transducing the GABA signal for improving water use efficiency, not ALMT12. We have therefore not performed this experiment.****

(3) AtALMT12 is involved in not only dark-induced stomatal closure but also abscisic acid (ABA)- and CO₂-induced stomatal closure (Meyer et al., 2010 Plant J; Sasaki et al., 2010 Plant Cell Physiol). Therefore, the authors need to check whether GABA can antagonize

stomatal closure induced by other stimuli such as ABA. This experiment would help well to understand mechanism and physiological function of guard cell GABA signaling at a deeper level.

****We have now included experiments examining the interaction of GABA with a variety of signals by applying the stimulus simultaneously with GABA (new Supplementary Figure 3). We agree this adds an extra dimension to the manuscript, albeit with the caveat that this was performed in epidermal peels. We have included additional data that shows when the ABA stimulus is small (low concentrations) GABA inhibits closure, but when the stimulus is high (high concentrations) then GABA no longer inhibits closure. It should be noted that ALMT12 is not the major anion channel associated with closure in guard cells. When the S-type anion channel is activated guard cells will close – this will occur at high ABA concentrations. The S-type anion channel can also be activated through a Ca²⁺-dependent mechanism. When we incubate stomata in 2 mM external Ca²⁺ GABA does not inhibit closure. For the referees' convenience we show that GABA can inhibit opening to low concentrations of coronatine (a toxin produced by *Pseudomonas syringae* that opens stomata), and closure via H₂O₂. We have not followed up on this collective data in this manuscript at this time as this would be out of the scope. However, this data does show that GABA can act as a universal brake on stomatal movement when the stimulus is not overpowering. In terms of drought signalling this equips GABA with a role in modifying movement akin to a faucet reducing stomatal opening and water loss from the leaf, whereas ABA acts more like a stop cock when the stimulus is significant enough and closes the stomata.****

(4) Considering the localization and membrane topology of ALMT9 and ALMT12, the two ALMTs might sense GABA at different cellular location. (Vacuole GABA targets ALMT9 and apoplastic GABA targets ALMT12?) Please mention and discuss this point.

****As we have removed the results around ALMT12 this statement is not really justifiable in the manuscript. Instead we have talked around this point in an expanded discussion from L401, which mentions that the GABA sensitive residues in ALMT9 are predicted to be cytosolic facing.****

Minor comment

(1) Please add scale bars to stomatal pictures of Supplementary Figure 3.

****As these pictures were illustrative only, we removed the images rather than inserting accurate scale bars in each.****

(2) L262: "ALMT" -> "ALMTs".

****Thank you – we have made this change.****

Reviewer #3 (Remarks to the Author):

GABA is a major metabolite in plants, which has been studied for decades. It is synthesized in the cytosol, and functions in primary carbon/nitrogen metabolism and in respiration.

Accordingly, the concentration of GABA in plant cells and in specific subcellular compartments is among the highest concentration of all amino acids, in the mM range. In addition, plant GABA was suggested to have functions as an osmoregulator (similar to proline) and pH regulation. GABA synthesis is enhanced by various stressors, mostly by activation of cytosolic glutamate decarboxylase (GAD), which is highly responsive to acidic pH, and to Ca²⁺/calmodulin at neutral pH. In Arabidopsis there are 5 genes encoding GAD enzymes, some with tissue specific expression. For example, GAD1 is mostly expressed in roots. A major question over the years has been the possible role of GABA as a signaling molecule, similar to its in animals. In recent years, the group of the corresponding author has shown that GABA can function as a negative regulator of anion flux through ALMT malate transporters. Yet, its role as a signaling molecule in stress responses has not been demonstrated.

In the submitted manuscript the authors suggest a role for GABA in controlling stomata aperture, transpiration and drought tolerance, via inhibition of plasma membrane and tonoplast-localized ALMT anion transporters. My review is divided in two parts. In the first part (A) I assess the results concerning the effects of GABA synthesis on stomata aperture, transpiration rate and drought tolerance. In the second part (B) I review the evidence concerning GABA function as a signaling molecule (rather than as a metabolite) in guard cells.

A. The results concerning GABA effects on stomata aperture in excised epidermal strips are consistent with the results obtained in intact plants regarding GABA-dependent transpiration rates and drought tolerance, demonstrating the importance of GABA. This is nicely shown by comparing wild type with gad2 mutants and to complementation lines expressing GAD2deltaC (presumably a constitutively active form of the enzyme) under the transcriptional regulation of a guard cell-specific promoter. I find this molecular – physiological part convincing, novel and worthy of publication. I do have a minor reservation concerning the authors conclusion that only GAD2 is operating in these processes. In lines 136-138 the authors describe the results presented in Supplementary Fig. 5 suggesting that transcript levels of GABA-related genes “were similar in wild type and gad2 lines”. I find this statement somewhat inaccurate, particularly on day 7 of the drought treatment. Similarly, the authors state that ABA-marker genes are expressed similarly in WT and gad2 mutants. This is also inaccurate, particularly on day 7 of the drought treatment of mutant gad2-1.

*****Thank you for this positive evaluation. We have corrected description of now modified in now Supplementary Figure 7. We have also added ABA concentration data for WT, gad2-1 and complementation lines under control conditions showing no differences between lines under controlled conditions (Supplementary Figure 8f).*****

B. Is GABA regulating stomata aperture, transpiration and drought tolerance by its role as a signaling molecule or as a major metabolite (e.g. in respiration, osmoregulation, etc.), or both? As the first step to assess if GABA functions as a signaling molecule, the authors identified ALMT anion transporters as possible targets of GABA in guard cells. The authors rely on their previous work showing that ALMT transporters contain a GABA-binding domain which negatively regulates anion transport when GABA binds. The authors here focused on ALMT9, a vacuolar anion channel essential for proper regulation of stomata opening, and on

the plasma membrane ALMT12, essential for proper stomata closing. Using mutants of both transporters, the authors nicely show the effects of GABA on the opening of stomata via ALMT9 and of closing stomata via ALMT12.

*Subsequently, in order to assess whether GABA affects stomatal conductance by binding to the expected site on ALMT9, the authors performed complementation of an *almt9* mutant with WT ALMT9 (Fig. 7e), or with a specific double mutation of ALMT9 (F243C/Y245C), which is expected to prevent the regulation by GABA, according to a previous publication of the corresponding author. However, the results are not perfect. The authors tested only two complementation lines (#1 and #2) for each construct. Of the two lines with the WT-expressed protein, complementation line ALMT9#1 did not restore the response to GABA (it should have) and complementation line ALMT9#2 restored an effect of GABA at a rather low statistical confidence ($p < 0.05$). Therefore, the fact that the two complementation lines with the mutations did not restore a response to GABA is not convincing enough for concluding an effect of the mutation on responsiveness to GABA. As this set of experiments is critical for drawing the main conclusion of the submitted manuscript, I believe that the authors should test more complementation lines (of WT and mutant) to provide more convincing data.*

****We absolutely agree. We repeated these experiments another 3 times and we consistently see that the double point mutant is not GABA sensitive. Furthermore, we performed GABA feeding into excised leaves of *almt9* and the complemented lines with ALMT9 and the double point mutant and measured gas exchange in real time (Figure. 8b-g). We are very pleased to report that we obtained a consistent result with the apertures that even more convincingly demonstrates that GABA acts on ALMT9, and via the putative GABA interaction site – also see answer 5 to reviewer 1.****

Another issue is related to the fact that recently ALMT1 from wheat was shown (by the corresponding author and colleagues) to function as a GABA transporter (Ramesh et al., 2018) and that mutation F213C (analogous to F243C in Arabidopsis ALM9) prevented GABA influx and efflux through the transporter (Ramesh et al., 2018). Hence, if the double mutation (F243C/Y245C) impedes GABA flux through the tonoplast and plasma membrane ALMTs, and consequently affects GABA subcellular compartmentalization in guard cells, it is possible that this effect is the actual driver of stomata aperture modulation under water deficit conditions. The authors should address this issue.

****The reviewer shares the interpretation that ALMT9 is transducing the GABA signal, but suggests that GABA flux through the channel may be the source of the phenotype affecting stomatal modulation rather than chloride movement. In a recent paper, Long, Tyerman and Gilliham, 2020, New Phytologist, 225:671-678 we were unable to detect electrogenic GABA flux through isolated wheat ALMT1. However, due to the above-named Ramesh et al 2018 two models exist for the GABA block of anion currents through ALMT1: 1) GABA associates whilst not traversing ALMT1 to inhibit anion flow – GABA traverses the membrane through other proteins; and, 2) GABA block of anion currents is due to ALMT1 changing its mode from anion permeation to GABA permeation. Regardless, the result is the same, anion currents are inhibited which would prevent cations and water from following – meaning swelling of the vacuole does not occur. We have been unable to definitively prove that GABA traverses ALMT1, or the other ALMTs, and besides there are other GABA permeable transport proteins in cells, so we suspect that our phenotypes are not explained due to

GABA compartmentation. Furthermore, the *almt9/ALMT9doublepointmutant* has stomata that open and close like wildtype (except for the lack of GABA block), unlike the *almt9* mutant, meaning that ion flow through the channel is required for normal opening. Regardless, we now invoke in the discussion in the new paragraph mentioned above from L403.****

Furthermore, the lack of an in vivo reporter for GABA, to determine real-time spatial and temporal GABA concentrations in different cellular compartment in the context of the stress response, makes it difficult to understand what really happens with GABA in guard cells. The authors determined whole leaf GABA levels in plant extracts but these cannot be interpreted as the levels of GABA in specific leaf cell types, or in specific cellular compartments.

****We agree entirely but this is experimentally impossible to address at the present time. Isolated guard cells undergo significant manipulation which will alter metabolite levels and so the validity of any such experiment is debatable. The reviewer is correct, the best way to address this is by developing a genetically encoded sensor. We have already mentioned (in response to Reviewer 1 – answer 10), we are developing an *in planta* sensor. This is in collaboration with Loren Looger (HHMI recently moved to UCSD), but this is still a couple of years away until we have a completely robust dataset. We have added this caveat in the discussion and will hopefully be able to address this in years to come when the sensor is available from L405. ‘This would be aided by the determination of GABA concentrations in different cell-types and compartments to further understand the co-ordination of GABA signalling across leaves and other organs, and this could be achieved through the deployment of novel GABA sensors as recently used in animal tissues.’****

Minor point

It is not clear to me why the authors chose to refer to the GABA signal as a “memory” signal. In order to determine that a substance acts like a memory signal, it would be necessary to determine its effects on subsequent stress responses, after the first stress has been alleviated for some time. In the submitted study the authors have not attempted this type of experiments, but rather studied the effects of GABA during a given stress response and immediately upon recovery from the stress. Therefore, I don’t find a justification to refer to this case as “memory”.

****We have refined the title to: GABA signalling modulates stomatal opening to enhance plant water use efficiency and drought resilience.****

Recommendation: The authors describe a novel mechanism of GABA controlling stomata aperture, transpiration and drought tolerance. These findings should be published in Nature Communication, pending revision.

****We thank the reviewer for their positive assessment, and hope our revisions are to their satisfaction.****

REVIEWER COMMENTS

Reviewer #1 (Remarks to the Author):

Xu et al have made major structural changes in this new version of the manuscript. It was first proposed that GABA was a negative regulator of stomatal movements. This would implicate that mutants lacking GABA would be more responsive to both stimulus-induced stomatal opening and closing. This hypothesis, however, could not be sustained by the phenotype observed in *gad2* mutants, which have more opened stomata and higher stomatal conductance.

Besides the problematic mechanism of action firstly proposed, the previous version of the manuscript did not contain much data on intact leaf gas exchange. The new version of the manuscript has fixed this issue by using data on GABA-fed intact leaves in multiple experiments. These new experiments greatly enhanced the quality of this research and further support a role of GABA as a regulator of stomatal opening.

1. In Figure 1A, the authors show that exogenous application of GABA in epidermal peels decreases dark-induced stomatal closure responses. However, in Figure 2C, leaves fed with GABA or control seem to similarly close their stomata in response to dark. The authors then suggest that GABA plays a "dominant effect" in modulating stomatal opening. This led them to suggest that the data on GABA-inhibition of stomatal closure through modulation of ALMT12 was "preliminary" and needs further research.

In this context, I am not fully convinced that "GABA antagonises stomatal movements" as stated in results in the title and this should be re-phased. This seems to be the case only for epidermal peels.

2. The authors have removed data on ALMT12, which does not fit the present model. It is recommended to not remove all data that does not "fit" the model and include some data. The authors already directly refer to future questions in this respect, but removing data does not seem appropriate and some relevant data pertaining to the discussion should be added back. Data that do not support a model are interesting and relevant.

3. The authors state they controlled the weight of the soil in the response letter: "We followed the standard protocols as outlined above (equalising soil weights, and watering to weight). This detail is now included in the methods."

But when I read the Methods the experimental procedure is not evident and the manuscript may be missing important information. The authors weighed pots with soil and then state on line 598 that pots were weighed "before seedling transferred from plates."

And on line 648:

"Fresh soil weight of the whole pot (M_{wet}) and dry soil weight after drying the soil (M_{dry}) at 105 °C for 3 d was measured using an Ohaus ARA520 Adventurer Balance. Gravimetric soil water content (θ_g) of the whole soil in the pots was calculated by an equation:..."

However, the Methods do not explain how pots were weighed during the drought treatment and whether water was added back to pots or not.

In Supplementary figure 6 soil water content is shown, but the authors should better describe the approach in Methods for the different time points. An important question here is whether either (a) soil/pot drying occurred at different rates in wildtype controls and tested plants or (b) whether

water was added back periodically to balance pots weights. Either approach can be interesting and valid, as long as the methods are evident, and the results are shown and interpreted correctly. For this purpose graphs as in Supplementary figure 6b need to show the different genotypes in the same panel for comparison and the conclusions should be adjusted dependent on these findings, meaning to the plants show different phenotypes during drought because (a) the soil has dried out more in one genotype or (b) because the plants withstand drought better at equal soil moisture?

Minor:

The term “dominant” is used in several contexts. It is suggested to reword this unless the genetic mutant context is meant.

It is worthwhile mentioning that the ability to perform new experiments has been reduced, in the context of a pandemic and in respect to social distancing. Nonetheless, the authors were able to experimentally address many of the concerns raised by the reviewers. This version of the manuscript proposes a new regulation mechanism during stomatal opening responses and a new player in this sophisticated biological process. Making these final revisions is recommended.

Reviewer #2 (Remarks to the Author):

The previous important work (Mekonnen et al., 2016, *Plant Science* 245: 25–34) has already reported the fundamental data showing that GABA functions as a regulator of stomatal movement and drought tolerance in plants. The major advance shown in this paper is that ALMTs (ALMT9) mediate the guard cell GABA signaling, but the mechanism is not fully evidenced. The reviewer agrees with the authors’ comment on technical issues regarding electrophysiological analysis of ALMT9. However, the authors need to check GABA-binding activity of ALMT9 and ALMT9 F243C/Y245C using a different approach (comment 3). More careful analysis of ALMT9 complement lines and ALMT9 F243C/Y245C subcellular localization (comment 2) is also required to justify the authors’ conclusion.

1) The authors have removed the data regarding ALMT12 to avoid some confusion, but the reviewer does not agree with the authors’ decision. Although the authors could not observe clear phenotype of *almt12* mutants with WUE and drought tolerance assays (possibly due to the functional redundancy with other ALMTs or slow-type anion channels), the previous data regarding ALMT12 is important and required to fully prove the physiological role of the GABA-ALMT interaction in guard cells.

2) Expression of the wildtype *ALMT9* rescued the *almt9-2* stomatal phenotype, but expression of GABA-insensitive ALMT9 (F243C/Y245C) did not (Fig. 8 and Suppl Fig. 15). In addition, the *ALMT9 F243C/Y245C* complement lines, but not wildtype *ALMT9* complement lines, showed greater steady state stomatal conductance than wildtype (Fig. 9). These are key results that support the authors’ model in which GABA modulates stomatal opening by regulating ALMT9 activity. However, it is also possible that the observed *ALMT9 F243C/Y245C* phenotype is due to the lower expression and/or mislocalization of the ALMT9 F243C/Y245C, not due to the lack of the GABA-binding activity. To confirm this, it is highly recommended to check (i) mRNA transcripts of *ALMT9 F243C/Y245C* and wildtype *ALMT9* in the complement lines and (ii) subcellular localization of ALMT9 F243C/Y245C.

3) Previously the author group used fluorescent muscimol-BODIPY to confirm that wheat TaALMT1 binds to GABA in planta and in the heterologous expression system (*Xenopus oocyte*) (Ramesh et al., 2015 *Nature Communications*). Similar experiments using isolated vacuoles from *Arabidopsis* or ALMT9-overexpressing tobacco (or *Xenopus oocytes*) would demonstrate whether F243/Y245 residues of ALMT9 are really responsible for the GABA binding.

4) The authors have removed guard cell images from supplemental figure 5, but the reviewer disagrees with the revision. The representative images had clearly visualized the response of kidney-shaped (dicot) and dumbbell-shaped (monocot) guard cells to GABA. Such figures would make the results easy to understand for other field researchers.

Reviewer #3 (Remarks to the Author):

The authors of the submitted manuscript investigated the mechanism underlying GABA regulation of stomata conductance and its relevance to plant survival under conditions of water deficiency. They report that GABA plays a role in limiting stomata opening under stress, and therefore is important for drought tolerance and enhanced water use efficiency. They further demonstrated this phenomenon in various plant species. Subsequently the authors investigated the mechanism underlying the GABA effect on stomata conductance in Arabidopsis. Based on elaborate molecular-genetic and physiological investigations, the authors revealed that GABA operates by binding to the ALMT9 vacuolar transporter. Compared to the original submission, in the revised manuscript the evidence for the role of GABA as a signaling molecule operating to restrict stomata opening under drought conditions through binding to specific residues in ALMT9 has improved and is convincing. Indeed, this is the major novel finding reported in the manuscript. This finding is consistent with their original work that revealed the binding site of GABA to wheat ALMT (Ramesh et al., 2015; Ref 18). In the revised manuscript, the authors also demonstrate the occurrence of cross-talk between GABA and other signaling mechanisms that are known to affect stomata conductance (e.g. ABA), although the mechanism of such cross talk has not been investigated.

It should be mentioned that the role of GABA in modulating stomata conductance and improving drought tolerance was reported earlier by Mekonnen et al (2016; Ref 29) but without addressing the underlying mechanism.

Regarding remaining open questions, the nature of the GABA signal including its concentration and spatial and temporal dynamics in response to drought and upon recovery from drought, remains obscure. There is yet no in vivo GABA reporter in plants. A recent GABA reporter has been developed for animal cells (Marvin et al., 2019; Ref. 53). The authors refer to this reporter in the revised discussion.

It also remains unclear how GABA binding to vacuolar ALMT9 affects stomata conductance, as the authors have not conducted electrophysiological experiments to characterize this phenomenon. The authors cannot rule out a direct effect of GABA on stomata conductance by compartmentation via ALMT9. The authors address this issue in the revised discussion.

Minor comments:

Line 179: "constitutively" is redundant.

Lines 247 – 249: Since the regulatory domain of GAD2 is known to be inhibitory, it should be rephrased to state that REMOVAL of this domain, rather than its occurrence, "is important in stimulating GABA production under drought in guard cells". Please cite a relevant reference(s).

Line 430: add "to" before "light and dark".

Response to *reviewer* delimited by **, **. Line numbers refer to pdf, not tracked changes word doc.

REVIEWER COMMENTS

Reviewer #1 (Remarks to the Author):

Xu et al have made major structural changes in this new version of the manuscript. It was first proposed that GABA was a negative regulator of stomatal movements. This would implicate that mutants lacking GABA would be more responsive to both stimulus-induced stomatal opening and closing. This hypothesis, however, could not be sustained by the phenotype observed in *gad2* mutants, which have more opened stomata and higher stomatal conductance.

Besides the problematic mechanism of action firstly proposed, the previous version of the manuscript did not contain much data on intact leaf gas exchange. The new version of the manuscript has fixed this issue by using data on GABA-fed intact leaves in multiple experiments. These new experiments greatly enhanced the quality of this research and further support a role of GABA as a regulator of stomatal opening.

1. In Figure 1A, the authors show that exogenous application of GABA in epidermal peels decreases dark-induced stomatal closure responses. However, in Figure 2C, leaves fed with GABA or control seem to similarly close their stomata in response to dark. The authors then suggest that GABA plays a “dominant effect” in modulating stomatal opening. This led them to suggest that the data on GABA-inhibition of stomatal closure through modulation of ALMT12 was “preliminary” and needs further research.

In this context, I am not fully convinced that “GABA antagonises stomatal movements” as stated in results in the title and this should be re-phased. This seems to be the case only for epidermal peels.

We agree that we could have been more specific, we have changed the title to: “GABA antagonises both stomatal opening and closure in epidermal peels, but only opening in leaf feeding experiments”.

2. The authors have removed data on ALMT12, which does not fit the present model. It is recommended to not remove all data that does not “fit” the model and include some data. The authors already directly refer to future questions in this respect, but removing data does not seem appropriate and some relevant data pertaining to the discussion should be added back. Data that do not support a model are interesting and relevant.

**We have added back in that GABA inhibition of epidermal strips is dependent upon ALMT12 (Suppl. Fig. 13), and that stomatal responsiveness to GABA is completely lost in *almt12 x almt9*

mutants (Suppl. Fig. 18).**

3. The authors state they controlled the weight of the soil in the response letter:

“We followed the standard protocols as outlined above (equalising soil weights, and watering to weight). This detail is now included in the methods.”

But when I read the Methods the experimental procedure is not evident and the manuscript may be missing important information. The authors weighed pots with soil and then state on line 598 that pots were weighed “before seedling transferred from plates.”

And on line 648:

“Fresh soil weight of the whole pot (M_{wet}) and dry soil weight after drying the soil (M_{dry}) at 105 °C for 3 d was measured using an Ohaus ARA520 Adventurer Balance. Gravimetric soil water content (ϑ_g) of the whole soil in the pots was calculated by an equation:...”

However, the Methods do not explain how pots were weighed during the drought treatment and whether water was added back to pots or not.

We have re-written the methods, and moved the soil and tissue weight assays to follow the growth conditions. We have specifically stated that pots were not rewatered during the assays.

In Supplementary figure 6 soil water content is shown, but the authors should better describe the approach in Methods for the different time points. An important question here is whether either (a) soil/pot drying occurred at different rates in wildtype controls and tested plants or (b) whether water was added back periodically to balance pots weights. Either approach can be interesting and valid, as long as the methods are evident, and the results are shown and interpreted correctly. For this purpose graphs as in Supplementary figure 6b need to show the different genotypes in the same panel for comparison and the conclusions should be adjusted dependent on these findings, meaning to the plants show different phenotypes during drought because (a) the soil has dried out more in one genotype or (b) because the plants withstand drought better at equal soil moisture?

**We did not rewater during the drought period, this was mentioned in the results (now L151) and now is clearly restated in the methods (L632) and in the figure legend of Suppl. Fig. 6.

Our physiological data clearly shows that both *gad2* stomatal aperture and stomatal conductance is greater than wildtype. This indicates that *gad2* soil dries out more quickly because the plants use soil water more quickly. Water use efficiency is a major trait impacted by GABA, and this is known to lead to greater drought resilience when water is limiting.

We have specifically stated that in the results (L144-148) and the implications within the discussion (associated with Fig. 10 & L452-560).**

Minor:

The term “dominant” is used in several contexts. It is suggested to reword this unless the genetic mutant context is meant.

****We have changed the word dominant to major.****

It is worthwhile mentioning that the ability to perform new experiments has been reduced, in the context of a pandemic and in respect to social distancing. Nonetheless, the authors were able to experimentally address many of the concerns raised by the reviewers. This version of the manuscript proposes a new regulation mechanism during stomatal opening responses and a new player in this sophisticated biological process. Making these final revisions is recommended.

****Many thanks for this. It has been and continues to be a very tough year for many around the world; we have done everything we can to focus on addressing the reviewer requests.****

Reviewer #2 (Remarks to the Author):

The previous important work (Mekonnen et al., 2016, Plant Science 245: 25–34) has already reported the fundamental data showing that GABA functions as a regulator of stomatal movement and drought tolerance in plants. The major advance shown in this paper is that ALMTs (ALMT9) mediate the guard cell GABA signaling, but the mechanism is not fully evidenced. The reviewer agrees with the authors’ comment on technical issues regarding electrophysiological analysis of ALMT9. However, the authors need to check GABA-binding activity of ALMT9 and ALMT9 F243C/Y245C using a different approach (comment 3). More careful analysis of ALMT9 complement lines and ALMT9 F243C/Y245C subcellular localization (comment 2) is also required to justify the authors’ conclusion.

****In response to the above points raised, we feel it is important that we provide the following clarifications as a context for our responses to the reviewers’ requests:**

- a) We respectfully disagree with the assessment that “previous important work (Mekonnen et al., 2016, Plant Science 245: 25–34) has already reported the fundamental data showing that GABA functions as a regulator of stomatal movement and drought tolerance in plants”. I do not wish to cast aspersions on others work, so I hope this isn’t taken the wrong way, but it is not possible to make such a strong statement or conclude this as a finding from the data presented in Mekonnen et al (2016). For instance, there are several basic issues including, but not limited to, the use only one mutant allele (which is also a complex mutant with developmental phenotypes including altered stomatal density), and the lack of standard genetic complementation. The 2016 paper provides a hint that GABA may affect water use (as have other papers),

but it falls short of providing convincing evidence that GABA is a signal.

- b) The major advances we show are more than just the role of ALMT9 in transducing GABA signals in stomata including:

Cytosolic GABA regulates stomatal aperture – and it does this by being a signal not as a metabolite or through developmental means.

GABA inhibits opening, rather than stimulating closure.

We show the effect can be guard cell specific and Ca^{2+} signaling is important in GABA stimulation by drought

We show for the first time that you can improve in WUE & drought tolerance, through modifying GABA content both by complementation in knockout plants and by using *GAD2* overexpression in wildtype over and above standard drought tolerance .

We have shown interaction of GABA with other signals such as ABA

That GABA regulates stomatal aperture in multiple plant species.

First *in planta* modulation of ALMT sensitivity to GABA

A clear link between GABA and a physiological output – which demonstrates the role of GABA as a plant signalling molecule

- c) We do not claim to have demonstrated that GABA directly modulates ALMT9 in this paper. In fact, GABA directly binding to ALMT has not been shown in any published work to date – it has only been inferred through electrophysiology and non-specific fluorescent assays.

Taken together we suggest that the extent of the novelty, significance and advance of our work may have not been fully apparent or appreciated.

We have made revisions in the manuscript addressing all five points suggested by the reviewer, which has involved additional experimentation.**

*1) The authors have removed the data regarding ALMT12 to avoid some confusion, but the reviewer does not agree with the authors' decision. Although the authors could not observe clear phenotype of *almt12* mutants with WUE and drought tolerance assays (possibly due to the functional redundancy with other ALMTs or slow-type anion channels), the previous data regarding ALMT12 is important and required to fully prove the physiological role of the GABA-ALMT interaction in guard cells.*

** As requested, we have added the data that shows ALMT12 is required for GABA to inhibit stomatal closure in epidermal peels and the complete GABA insensitivity of *almt9xalmt12* knockouts (Suppl. Fig. 13 and 18).**

2) Expression of the wildtype ALMT9 rescued the almt9-2 stomatal phenotype, but expression of GABA-insensitive ALMT9 (F243C/Y245C) did not (Fig. 8 and Suppl Fig. 15). In addition, the ALMT9 F243C/Y245C complement lines, but not wildtype ALMT9 complement lines, showed greater steady state stomatal conductance than wildtype (Fig. 9). These are key results that support the authors' model in which GABA modulates stomatal opening by regulating ALMT9 activity. However, it is also possible that the observed ALMT9 F243C/Y245C phenotype is due to the lower expression and/or mislocalization of the ALMT9 F243C/Y245C, not due to the lack of the GABA-binding activity. To confirm this, it is highly recommended to check (i) mRNA transcripts of ALMT9 F243C/Y245C and wildtype ALMT9 in the complement lines and (ii) subcellular localization of ALMT9 F243C/Y245C.

**We agree that this adds important information. As requested, we now provide further data to show that:

- i) ALMT9-GFP and the ALMT9-GFP point mutant have the same localization (being present on the tonoplast of complemented plants (New Supp Fig 16c, d); and,
- ii) that the expression of *ALMT9 F243C/Y245C* and wildtype *ALMT9* are similar in the complement lines (New Supp Fig 16a, b).

Furthermore, we have provided additional data that shows that the point mutant expressing plants have a greater stomatal aperture than those expressing ALMT-GFP (New Supp Fig 17j).

Collectively, this is convincing evidence that the point mutations do not alter the localization of ALMT9-GFP, that the mutant protein is functional and leads to greater stomatal opening compared to ALMT9-GFP, and the GABA phenotype is not related to expression level differences.

This marries with the data already in the manuscript that shows that the point mutation plants have greater stomatal conductance (Fig 9). As such, this data strengthens our conclusion that the F243C/Y245C mutations within ALMT9 prevents the GABA signal being transduced but leads to an otherwise functional protein.**

3) Previously the author group used fluorescent muscimol-BODIPY to confirm that wheat TaALMT1 binds to GABA in planta and in the heterologous expression system (Xenopus oocyte) (Ramesh et al., 2015 Nature Communications). Similar experiments using isolated vacuoles from Arabidopsis or ALMT9-overexpressing tobacco (or Xenopus oocytes) would demonstrate whether F243/Y245 residues of ALMT9 are really responsible for the GABA binding.

**We claim that F243/Y245 are important in transducing the GABA signal. We do not claim that we show these amino acid residues are important in GABA binding.

As stated above, no-one has yet shown definitively that ALMTs bind GABA.

The experiments suggested above would be technically very challenging and would be extremely unlikely to render a conclusive result for many technical reasons, including lack of functionality of ALMT9 in oocytes, fluorescence interference in protoplasts, and the need to find an alternative assay to show that muscimol is specifically binding to ALMT9.

To definitively demonstrate GABA binding would be a substantial undertaking, would be a major standalone study and would require many months of experimentation, and many additional figures to an already data dense and impactful manuscript.

Taking all this into consideration we have amended our results and discussion sections to prevent the misconception that we show that GABA binding is affected by the amino acid residue mutations and importantly state that no-one has yet definitively shown GABA binding for ALMTs. In fact, we now make the argument that our data adds further impetus for the need to show whether or not ALMTs bind GABA or whether the GABA response is affected by these mutations via a non-direct path (L422-436).

We thank the review for stimulating this change to the manuscript, we believe it has led to a more compelling narrative in the discussion and it has further increased the downstream impact of our work.**

4) The authors have removed guard cell images from supplemental figure 5, but the reviewer disagrees with the revision. The representative images had clearly visualized the response of kidney-shaped (dicot) and dumbbell-shaped (monocot) guard cells to GABA. Such figures would make the results easy to understand for other field researchers.

Thank you for this suggestion. We have added these images back in.

Reviewer #3 (Remarks to the Author):

The authors of the submitted manuscript investigated the mechanism underlying GABA regulation of stomata conductance and its relevance to plant survival under conditions of water deficiency. They report that GABA plays a role in limiting stomata opening under stress, and therefore is important for drought tolerance and enhanced water use efficiency. They further demonstrated this phenomenon in various plant species. Subsequently the authors investigated the mechanism underlying the GABA effect on stomata conductance in Arabidopsis. Based on elaborate molecular-genetic and physiological investigations, the authors revealed that GABA operates by binding to the ALMT9 vacuolar transporter. Compared to the original submission, in the revised manuscript the evidence for the role of

GABA as a signaling molecule operating to restrict stomata opening under drought conditions through binding to specific residues in ALMT9 has improved and is convincing. Indeed, this is the major novel finding reported in the manuscript. This finding is consistent with their original work that revealed the binding site of GABA to wheat ALMT (Ramesh et al., 2015; Ref 18). In the revised manuscript, the authors also demonstrate the occurrence of cross-talk between GABA and other signaling mechanisms that are known to affect stomata conductance (e.g. ABA), although the mechanism of such cross talk has not been investigated.

It should be mentioned that the role of GABA in modulating stomata conductance and improving drought tolerance was reported earlier by Mekonnen et al (2016; Ref 29) but without addressing the underlying mechanism.

Regarding remaining open questions, the nature of the GABA signal including its concentration and spatial and temporal dynamics in response to drought and upon recovery from drought, remains obscure. There is yet no in vivo GABA reporter in plants. A recent GABA reporter has been developed for animal cells (Marvin et al., 2019; Ref. 53). The authors refer to this reporter in the revised discussion.

It also remains unclear how GABA binding to vacuolar ALMT9 affects stomata conductance, as the authors have not conducted electrophysiological experiments to characterize this phenomenon. The authors cannot rule out a direct effect of GABA on stomata conductance by compartmentation via ALMT9. The authors address this issue in the revised discussion.

****We thank the reviewer for the positive review of our manuscript.****

Minor comments:

Line 179: “constitutively” is redundant.

****We agree and have removed this word.****

Lines 247 – 249: Since the regulatory domain of GAD2 is known to be inhibitory, it should be rephrased to stated that REMOVAL of this domain, rather than its occurrence, “is important in stimulating GABA production under drought in guard cells”. Please cite a relevant reference(s).

****We agree that this is an ambiguous statement. We have rephrased this to say “This suggests that activation of GAD2 via its regulatory domain is important in stimulating GABA production under drought in guard cells.” We have cited the relevant reference (L253).**

Line 430: add “to” before “light and dark”.

****Thank you. We have made this change.****

REVIEWERS' COMMENTS

Reviewer #2 (Remarks to the Author):

In accordance with the reviewers' comments, the authors revised the manuscript. I still have two comments.

(1) It is very strange for me (and maybe also for others in this research field) that this manuscript did not perform any electrophysiological analysis of GABA regulation of ALMT9 and ALMT12. It would be better if the manuscript states in the text that the authors tried electrophysiological analysis of ALMT9 but the analysis was not working at least under the tested condition (as mentioned in the previous response letter).

(2) Like ALMT9, does ALMT12 also have the putative GABA binding residue predicted to face the cytosol? Maybe yes, but please provide the information in the manuscript.

Changes delimited with **,**

REVIEWERS' COMMENTS

Reviewer #2 (Remarks to the Author):

In accordance with the reviewers' comments, the authors revised the manuscript. I still have two comments.

(1) It is very strange for me (and maybe also for others in this research field) that this manuscript did not perform any electrophysiological analysis of GABA regulation of ALMT9 and ALMT12. It would be better if the manuscript states in the text that the authors tried electrophysiological analysis of ALMT9 but the analysis was not working at least under the tested condition (as mentioned in the previous response letter).

****We have added this to the manuscript now. L332 (including new reference).****

(2) Like ALMT9, does ALMT12 also have the putative GABA binding residue predicted to face the cytosol? Maybe yes, but please provide the information in the manuscript.

****Added...L296.****